# Amplification of a broad transcriptional program by a common factor triggers the meiotic cell cycle in mice

**Mina L Kojima[1,2], Dirk G de Rooij[1], David C Page[1,2,3]\***

[1]Whitehead Institute, Cambridge, United States; [2]Department of Biology, Massachusetts Institute of Technology, Cambridge, United States; [3]Howard Hughes Medical Institute, Whitehead Institute, Cambridge, United States

**Abstract** The germ line provides the cellular link between generations of multicellular organisms, its cells entering the meiotic cell cycle only once each generation. However, the mechanisms governing this initiation of meiosis remain poorly understood. Here, we examined cells undergoing meiotic initiation in mice, and we found that initiation involves the dramatic upregulation of a transcriptional network of thousands of genes whose expression is not limited to meiosis. This broad gene expression program is directly upregulated by STRA8, encoded by a germ cell-specific gene required for meiotic initiation. STRA8 binds its own promoter and those of thousands of other genes, including meiotic prophase genes, factors mediating DNA replication and the G1-S cell-cycle transition, and genes that promote the lengthy prophase unique to meiosis I. We conclude that, in mice, the robust amplification of this extraordinarily broad transcription program by a common factor triggers initiation of meiosis.
DOI: https://doi.org/10.7554/eLife.43738.001

**\*For correspondence:**
dcpage@wi.mit.edu

**Competing interests:** The authors declare that no competing interests exist.

## Introduction

A key feature of sexual reproduction is meiosis, a specialized cell cycle in which one round of DNA replication precedes two rounds of chromosome segregation to produce haploid gametes. In most organisms, meiotic chromosome segregation depends on the pairing, synapsis, and crossing over of homologous chromosomes during prophase of meiosis I. These chromosomal events are generally conserved across eukaryotes and have been extensively studied (*Handel and Schimenti, 2010*).

Despite being a critical pivot point in the life cycle, *meiotic initiation* — the decision to embark on the one and only one meiotic program per generation — has been less studied, perhaps because the regulation of meiotic initiation is less conserved (*Kimble, 2011*). Because dissecting this transition requires access to cells on the cusp of meiosis, meiotic initiation has been studied most in budding yeast, which can be induced to undergo synchronous meiotic entry; there the transcription factor Ime1 upregulates meiotic and DNA-replication genes (*Kassir et al., 1988*; *Smith et al., 1990*; *van Werven and Amon, 2011*). In multicellular organisms with a segregated germ line, cells entering meiosis are difficult to access for detailed molecular study; they are found only within gonads, surrounded by both somatic cells and germ cells at other developmental stages. However, with recent tools enabling purification of large numbers of the specific cell type initiating meiosis, the male mouse now represents a tractable model for studying meiotic initiation in vivo (*Hogarth et al., 2013*; *Romer et al., 2018*). We can use pure populations of meiosis-initiating cells for biochemical assays to determine how germ cells transition to the meiotic cell cycle.

For the study of meiotic initiation in multicellular organisms, the mouse also has the advantage of an existing genetic model with clear defects starting from the very first steps of meiosis, including meiotic S phase, which begins in the middle of the preleptotene stage, and the initiation of homolog

pairing in the leptotene stage. In mice, the germ cell-specific gene *Stimulated by retinoic acid 8* (*Stra8*) is induced by rising retinoic acid levels at the start of meiosis, resulting in *Stra8* expression from the middle of the preleptotene stage through early in the leptotene stage (*Bowles et al., 2006*; *Endo et al., 2015*; *Koubova et al., 2006*; *Oulad-Abdelghani et al., 1996*; *Zhou et al., 2008*). Genetic studies have revealed *Stra8*'s key role in meiotic initiation. First, proper meiotic S phase depends on *Stra8*: on the C57BL/6 background, *Stra8*-deficient germ cells in females show no evidence of meiotic DNA replication; in males, they reach the preleptotene stage and show signs of DNA replication but do not exhibit the hallmark sign of meiotic S phase, which is loading of the meiotic cohesin REC8 (*Anderson et al., 2008*; *Baltus et al., 2006*; *Dokshin et al., 2013*). Furthermore, *Stra8*-deficient cells do not robustly express meiotic factors, or progress to the leptotene stage and begin the chromosomal events of meiosis (*Anderson et al., 2008*; *Baltus et al., 2006*; *Soh et al., 2015*). These genetic studies of *Stra8* deficiency suggest that the decision to initiate meiosis occurs in the middle of the preleptotene stage, upstream of meiotic S phase. [Note that the strict *Stra8* requirement for male meiotic initiation was not observed in mice of mixed genetic background, in which *Stra8*-deficient male germ cells initiated meiosis but could not complete meiotic prophase I (*Mark et al., 2008*).]

Curiously, however, genes involved in the chromosomal events of meiotic prophase I are often expressed and translated long before initiation of the meiotic cell cycle, in diverse species (*Christophorou et al., 2013*; *Jan et al., 2017*; *Pasierbek et al., 2001*; *Soh et al., 2015*). For example, REC8 and synaptonemal complex proteins are expressed in mouse spermatogonia, which undertake several mitotic divisions before reaching the preleptotene stage (*Evans et al., 2014*; *Wang et al., 2001*). This early expression of meiotic genes must be reconciled with the seemingly contradictory observation that meiosis is only initiated during the preleptotene stage.

Here, we isolate preleptotene stage germ cells — the cells initiating meiosis — to study how meiosis is triggered. We find that meiotic initiation coincides with upregulation of thousands of transcripts. Remarkably, this large ensemble of genes is upregulated by a common factor, STRA8, which we demonstrate is a transcriptional activator that directly binds their promoters. The STRA8-activated program of at least 1,351 genes includes not only canonical meiotic prophase genes but also many genes that have been extensively studied in the context of the mitotic cell cycle, including those critical for the G1-S transition and for DNA replication. We suggest that the robust amplification of such genes, together with factors that maintain the meiosis-specific extended prophase, triggers the one meiotic cell cycle in each generation.

## Results

### Transcriptional changes at the start of the meiotic cell cycle

To better understand how mouse germ cells initiate meiosis, we sought to identify transcriptional changes occurring at meiotic initiation in vivo. For this, we isolated male germ cells entering meiosis using developmental synchronization of spermatogenesis (*Romer et al., 2018*). By chemically modulating the levels of retinoic acid, which is required for spermatogonial differentiation (*Endo et al., 2015*; *van Pelt and de Rooij, 1990*), we can induce synchronous progression of germ cells through spermatogenesis and collect testes enriched for preleptotene cells (*Figure 1A*) (*Hogarth et al., 2013*; *Romer et al., 2018*). By histologically 'staging' a small tissue biopsy, we verified enrichment of desired cell types (*Romer et al., 2018*) and found that, in properly synchronized and staged ('2S') testes, preleptotene cells expressing the meiotic initiation marker STRA8 account for ~44% of all cells; they comprise only an estimated ~1.4% of cells in unperturbed adult testes (*Figure 1B,C*). Synchronization thus yields 31-fold enrichment in vivo of meiosis-initiating cells, enabling biochemical analyses of meiotic initiation with improved signal and decreased background noise. Importantly, preleptotene-enriched samples are not contaminated with any later stage germ cells, which might complicate efforts to understand meiotic initiation. By adding a sorting step to the 2S protocol — together called '3S' — we can isolate preleptotene cells with >90% purity (*Figure 1D*) (*Romer et al., 2018*).

We utilized *Stra8*-deficient mice as a genetic tool to reveal changes associated with meiotic initiation. Because cells deficient for *Stra8* reach the preleptotene stage but fail to initiate meiosis (*Anderson et al., 2008*), we can use RNA-seq to characterize the expression profiles of preleptotene

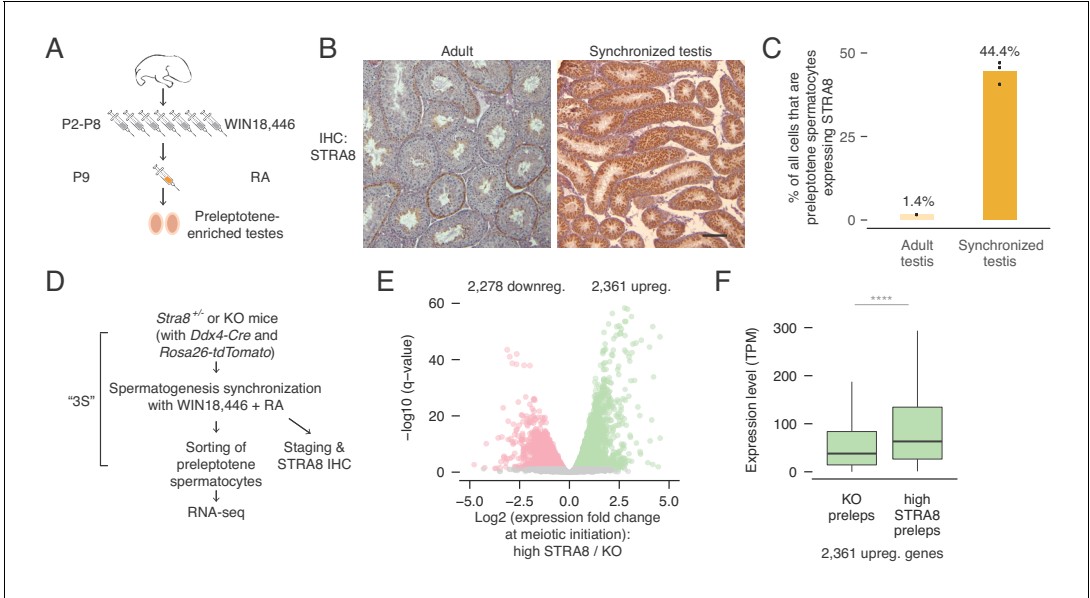

**Figure 1.** Transcriptional changes in the preleptotene stage population at meiotic initiation. (**A**) Schematic for synchronization of spermatogenesis by WIN18,446 and retinoic acid (RA) to enrich for preleptotene cells. (**B**) STRA8 immunohistochemistry (IHC) of an unperturbed adult testis, left, and a synchronized and staged (2S) testis, right. The synchronized testis is highly enriched for meiosis-initiating preleptotene spermatocytes that express STRA8. Scale bar = 100 μm. (**C**) Percent of all cells that are STRA8-expressing preleptotene cells, in adult testes or in 2S testes. Synchronization results in a 31-fold increase in the proportion of these cells. (**D**) Schematic for collection of pure preleptotene populations by synchronization, staging, and sorting (3S) for RNA-seq. Germ-cell lineage tracing allows for sorting of pure preleptotene populations from *Stra8*[+/-] or *Stra8*-deficient ('KO') testes. See *Figure 1—figure supplement 1* for cell sorting schematic. (**E**) Transcriptional changes associated with meiotic initiation. Volcano plot represents gene expression changes between *Stra8*[+/-] preleptotene cells with high levels of STRA8 (high STRA8) and *Stra8* KO preleptotene cells. Genes that are significantly (FDR < 0.05) downregulated or upregulated at meiotic initiation are in pink or green, respectively. Shown are 12,545 genes expressed in at least one preleptotene sample at transcripts per million (TPM) level ≥ 1. (**F**) Expression levels of genes upregulated at meiotic initiation, in KO and high-STRA8 preleptotene cells. Boxplots show sample medians and interquartile ranges (IQRs), with whiskers extending no more than 1.5 × IQR and outliers suppressed. ****$p < 2.2 \times 10^{-16}$, one-tailed Mann-Whitney *U* test. See *Figure 1—source data 1*.

DOI: https://doi.org/10.7554/eLife.43738.002

The following source data and figure supplements are available for figure 1:

**Source data 1.** Source data for RNA-seq analyses.
DOI: https://doi.org/10.7554/eLife.43738.005
**Figure supplement 1.** Gating strategy for sorting preleptotene cells.
DOI: https://doi.org/10.7554/eLife.43738.003
**Figure supplement 2.** RNA-seq analysis to identify transcriptional changes at meiotic initiation.
DOI: https://doi.org/10.7554/eLife.43738.004

cells with and without *Stra8* to identify gene expression changes at meiotic initiation. To do this, we purified preleptotene cells from *Stra8*-deficient and *Stra8*[+/-] testes using the 3S method (*Figure 1D* and *Figure 1—figure supplement 1*) (*Romer et al., 2018*); preleptotene cells from *Stra8*[+/-] testes were further categorized as displaying 'low STRA8' (early preleptotene stage, directly before meiotic initiation) or 'high STRA8' levels (mid or late preleptotene stage, after meiotic initiation) (*Figure 1D* and *Figure 1—figure supplement 2A*). In total, we profiled three biological replicates of *Stra8*-deficient, two of 'low-STRA8,' and two of 'high-STRA8' preleptotene samples. STRA8 protein level differences in the preleptotene samples paralleled *Stra8* mRNA level differences (average expression in 'low-STRA8' samples = 275 transcripts per million [TPM], and in 'high-STRA8' samples = 1,173 TPM; *Figure 1—figure supplement 2B*). Altogether, 12,545 genes were expressed at greater than one TPM in at least one preleptotene sample (*Supplementary file 1*).

To identify transcriptional changes at meiotic initiation, we first identified genes differentially expressed between high-STRA8 and *Stra8*-deficient preleptotene cells. The initiation of meiosis coincided with profound changes in transcript levels within the preleptotene population. Among expressed genes, 2,361 genes were upregulated, showing significantly higher expression in high-

STRA8 cells compared to *Stra8*-deficient preleptotene cells; another 2,278 genes were significantly downregulated (FDR < 0.05; *Figure 1E*; *Supplementary files 1* and *2*). We verified that these changes were due to differences in developmental progression through the preleptotene stage by comparing the genes' expression in *Stra8*$^{+/-}$ preleptotene samples with low and high STRA8 levels (*Supplementary file 2*). Overall, expression fold-changes in the low-STRA8/high-STRA8 comparison strongly correlated with expression fold-changes in the *Stra8*-deficient/high-STRA8 comparison (Pearson's *r*, 0.70; *Figure 1—figure supplement 2E*). In addition, of the genes up or downregulated at meiotic initiation, 93.9% and 89.2% were also up or downregulated, respectively, between *Stra8*$^{+/-}$ preleptotene cells expressing low or high levels of STRA8 (*Figure 1—figure supplement 2F*). These results confirm dramatic changes in the preleptotene stage transcriptome at meiotic initiation.

To further characterize changes in the preleptotene cells, we asked how many of the upregulated genes were 'off' in the absence of meiotic initiation. To our surprise, we discovered that 97% of upregulated genes were expressed (at TPM levels greater than one; median expression level 38 TPM) in *Stra8*-deficient preleptotene cells that had not initiated meiosis (*Figure 1F*). This suggests that transcriptional upregulation at meiotic initiation mostly entails amplification of gene expression, rather than 'turning on' genes that were previously not expressed.

## Identifying the targets of a meiotic initiation factor

We sought to determine whether the transcriptional changes we discovered at meiotic initiation were direct consequences of STRA8 action. STRA8 has been hypothesized to be a transcription factor (*Baltus et al., 2006*; *Soh et al., 2015*) and displays transcriptional activity in vitro; in cultured HEK293 cells, mouse STRA8 fused to a GAL4 DNA binding domain will activate transcription of a reporter gene (*Tedesco et al., 2009*). However, whether STRA8 directly regulates transcription in vivo remains unknown. To facilitate biochemical analyses of STRA8, we generated an N-terminal epitope-tagged allele, which we verified encodes a functional STRA8 protein (*Stra8*$^{FLAG}$; *Figure 2—figure supplement 1*) that is readily detected by the FLAG antibody (*Figure 2A*). Both male and female *Stra8*$^{FLAG/FLAG}$ mice were fully fertile and produced viable offspring, and their gonads displayed normal histology (*Figure 2—figure supplement 1D,F*).

Our N-terminal epitope tags only the longer of two STRA8 isoforms, which we found is the one critical for meiotic initiation. We generated a *Stra8* allele that selectively eliminates the longer isoform (*Figure 2—figure supplement 1A–D*). Mice homozygous for this new allele retained expression of the shorter STRA8 isoform but phenocopied *Stra8*-deficient mice, which express neither isoform (*Figure 2—figure supplement 2E–I*) (*Anderson et al., 2008*; *Baltus et al., 2006*). Although STRA8 is both nuclear and cytoplasmic in wild-type preleptotene cells, the protein was predominantly cytoplasmic in mice expressing only the short isoform (*Figure 2—figure supplement 3A,B*). Thus, the first 111 amino acids, unique to the long isoform, are required for STRA8's function and nuclear localization in vivo (see Appendix 1). This critical region contains the putative bHLH domain, suggesting that STRA8 regulates transcription.

To test whether STRA8 regulates transcription through direct binding of genomic regulatory elements, we performed ChIP-seq, using the FLAG antibody to immunoprecipitate chromatin from *Stra8*$^{FLAG/FLAG}$ testes (*Figure 2B*). For ChIP, we used 2S testes in which >90% of tubules contained STRA8-expressing preleptotene cells (*Figure 2—figure supplement 4A*). We sequenced three biological replicates, each representing chromatin from one mouse. We also sequenced three FLAG ChIP replicates using chromatin from 2S wild-type (*Stra8*$^{+/+}$) testes, which controls for non-specific antibody binding. [Note that STRA8 is also expressed in type A spermatogonia, where it plays a role in spermatogonial differentiation (*Endo et al., 2015*). However, our 2S testis samples were carefully selected following IHC and staging to ensure that most of the STRA8 ChIP-seq signal comes from cells at meiotic initiation. Of all cells that express STRA8 in our in our 2S samples, ~88% are initiating meiosis and ~12% are spermatogonia. As seen in *Figure 2A*, STRA8 levels are also much higher in preleptotene cells than in spermatogonia. The cellular compositions of 2S samples used for ChIP-seq are described in *Figure 2—figure supplement 4B*.]

Our ChIP-seq analyses reveal that, in preleptotene cells, STRA8 binds genomic regulatory regions. Peaks of heightened read density, reflecting likely STRA8 binding sites, were abundant in *Stra8*$^{FLAG/FLAG}$ samples but not in wild-type controls (*Figure 2—figure supplement 4C*). 83.0% of such peaks were within 1 kb of an annotated transcription start site (TSS), and across all genes,

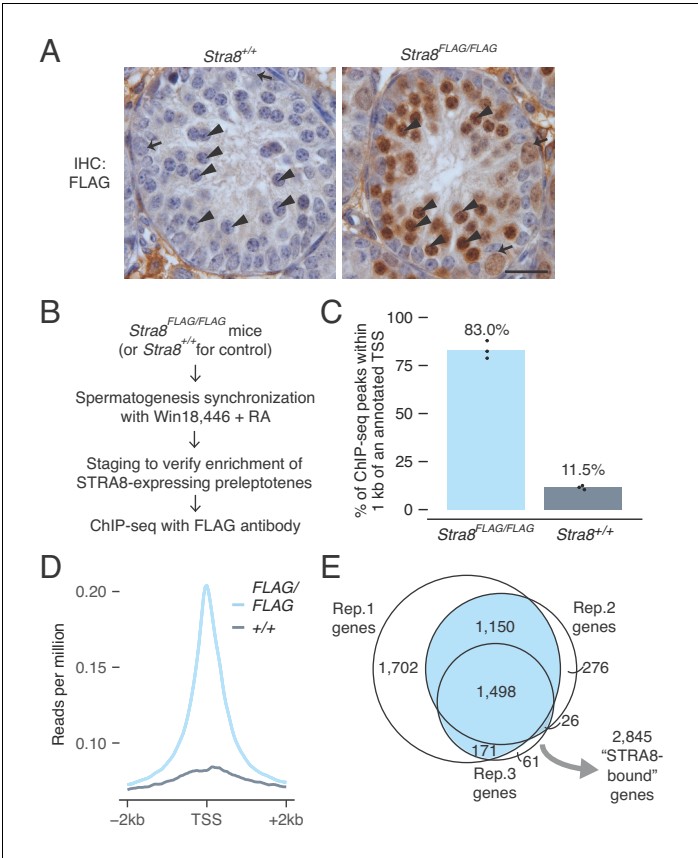

**Figure 2.** Identification of genes bound at meiotic initiation by STRA8. (**A**) FLAG IHC of preleptotene-enriched tubules in wild-type or *Stra8^FLAG/FLAG^* 2S testes. Preleptotene germ cells are marked by arrowheads, and spermatogonia by arrows. Scale bar = 20 μm. (**B**) Schematic for STRA8 ChIP-seq experiments using preleptotene-enriched testes. (**C**) Percent of ChIP-seq peaks at protein-coding gene promoters, defined as the window within 1 kb of annotated transcription start sites (TSS). Bar heights represent the average of three replicates, and dots indicate values in individual replicates. For comparison, 2.3% of the mouse genome lies within these regions. See also *Figure 2—figure supplement 4*. (**D**) Average *Stra8^FLAG/FLAG^* and *Stra8^+/+^* ChIP seq profiles over the TSS of genes. Sequencing reads were pooled from three ChIP replicates. See *Figure 2—figure supplement 4* for profiles of individual replicates. (**E**) Overlap of promoter-bound genes identified in *Stra8^FLAG/FLAG^* ChIP-seq replicates. Genes identified in at least two of the three replicates are considered 'STRA8-bound genes'. See *Figure 2—source data 1*.

DOI: https://doi.org/10.7554/eLife.43738.006

The following source data and figure supplements are available for figure 2:

**Source data 1.** Source data for STRA8 binding at promoters.

DOI: https://doi.org/10.7554/eLife.43738.011

**Figure supplement 1.** Generation and characterization of the *Stra8^FLAG^* knock-in mouse.

DOI: https://doi.org/10.7554/eLife.43738.007

**Figure supplement 2.** The longer STRA8 protein isoform is required for meiotic initiation in both males and females.

DOI: https://doi.org/10.7554/eLife.43738.008

**Figure supplement 3.** Nuclear localization of STRA8 requires its N terminus.

DOI: https://doi.org/10.7554/eLife.43738.009

**Figure supplement 4.** ChIP-seq analysis to determine sites of STRA8 binding.

DOI: https://doi.org/10.7554/eLife.43738.010

ChIP-seq read density was highest at the TSS (*Figure 2C,D* and *Figure 2—figure supplement 4C, D*), suggesting that STRA8 binds primarily at promoters. This promoter enrichment was not a consequence of non-specific antibody binding, because only 11.5% of ChIP-seq peaks in wild-type control were at promoters, and TSS read density was low in these controls (*Figure 2C,D* and *Figure 2—figure supplement 4C–E*). We explored whether the remaining STRA8 binding sites (in *Stra8*$^{FLAG/FLAG}$ samples) could be at enhancers, which are also sites of transcriptional regulation. Here we cross-referenced ENCODE consortium adult testis ChIP-seq data for H3K4me1, an enhancer-associated histone modification (*Heintzman et al., 2009*). Only 4.8% of STRA8 binding sites overlapped with H3K4me1-marked regions not at gene promoters (*Figure 2—figure supplement 4C*). Altogether, these results reveal that STRA8 primarily acts as a promoter-proximal transcriptional regulator.

To define the scope of the STRA8-regulated transcriptional program, we catalogued the STRA8-bound genes. We identified 4,884 protein-coding genes with a STRA8 peak (*Figure 2E* and *Figure 2—figure supplement 4C*). Even genes that were bound in only one replicate are likely true STRA8 targets; their promoters display ChIP-seq signal (*Figure 2—figure supplement 4G*). However, to ensure robust conclusions regarding the effects of STRA8 binding, we focus our analyses on target genes that overlapped between ChIP-seq biological replicates ($p<1\times10^{-320}$ for all pairwise overlaps between replicates, one-tailed hypergeometric tests). We refer to the 2,845 genes with peaks in at least two replicates as 'STRA8-bound genes' (*Figure 2E*), of which 2,809 were expressed in preleptotene cells.

## A broad gene expression program upregulated at meiotic initiation is directly bound by a common factor

To determine whether transcriptional changes at meiotic initiation are directly regulated by STRA8, we asked whether STRA8 binding could account for the up- and downregulation of gene expression at meiotic initiation; we tested for significant overlap between the differentially expressed genes and the STRA8-bound genes. Genes upregulated at meiotic initiation significantly overlapped with the upregulated STRA8-bound genes ($p<6.8\times10^{-322}$, one-tailed hypergeometric test; *Figure 3A*): 1,351 of the 2,361 upregulated genes (57.2%) were STRA8-bound (*Supplementary file 1*). If we relax our ChIP-seq criterion to include all genes with a STRA8 peak in any of the ChIP-seq replicates, 72.8% of upregulated genes may be directly activated by STRA8. Conversely, genes downregulated at meiotic initiation were significantly depleted for STRA8 binding ($p<7.5\times10^{-99}$, one tailed hypergeometric test; *Figure 3B*). Altogether, these analyses reveal a strong association between the program upregulated at meiotic initiation and STRA8 binding.

Inverting the analysis, our data provide evidence that STRA8 functions primarily as a transcriptional activator. Of expressed STRA8-bound genes that were differentially expressed at meiotic initiation, 89% were upregulated. In contrast, among the set of differentially expressed genes that were not STRA8-bound, only 32.4% were upregulated (*Figure 3C,D*). STRA8-bound genes are thus strongly biased towards upregulation at the start of meiosis (odds ratio = 17.1, 95% confidence interval 14.3–20.6, $p<2.2\times10^{-16}$, two-tailed Fisher's exact test; *Figure 3D*). In addition, expressed STRA8-bound genes, compared to a control set of other expressed genes, were more significantly upregulated ($p<2.2\times10^{-16}$, one-tailed Mann Whitney *U*; *Figure 3E*).

Our data reveal the breadth of the gene expression program robustly upregulated at meiotic initiation by a common factor: as many as 3128 genes may be directly activated by STRA8, if we use relaxed ChIP-seq and RNA-seq criteria (*Figure 3—figure supplement 1*). However, we focused subsequent analyses on a stringently validated set of 1,351 genes that displayed STRA8 binding in at least two of the three ChIP-seq replicates, with significantly higher expression in high-STRA8 compared to *Stra8*-deficient preleptotene cells (*Figure 3A*). 96% of these genes were also upregulated between low-STRA8 and high-STRA8 preleptotene cells (*Figure 3F*), corroborating their upregulation at meiotic initiation. We consider these 1,351 genes to be the set of 'STRA8-activated genes'.

## Mechanisms of gene activation at meiotic initiation

We sought to better understand how gene expression is upregulated at meiotic initiation. First, we asked whether STRA8 turns on genes that are otherwise silent. We found that 98% of STRA8-bound genes were expressed (at TPM levels greater than one; median expression level 104 TPM) even in *Stra8*-deficient cells that had not initiated meiosis (*Figure 4A*). Thus, STRA8 likely binds open

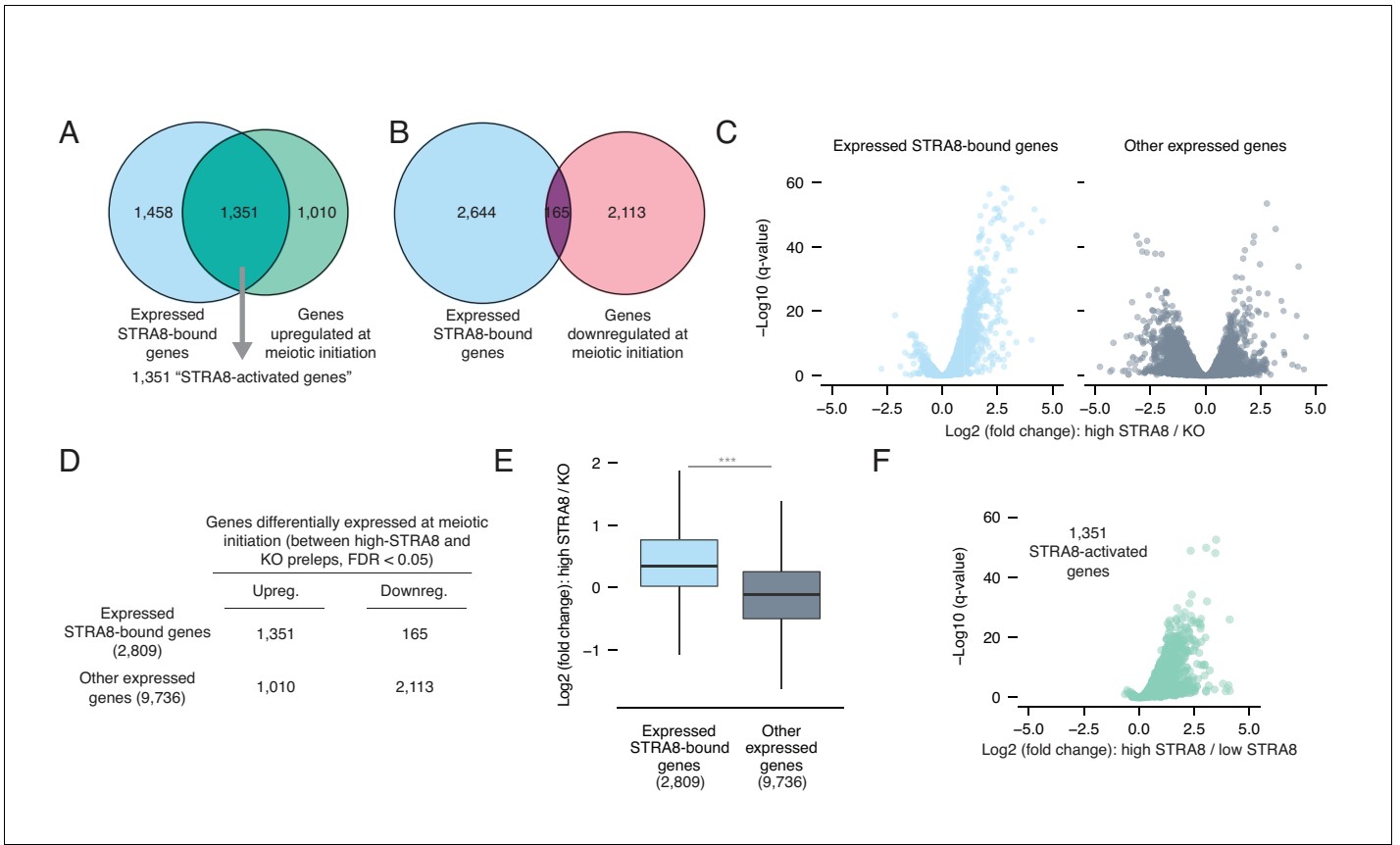

**Figure 3.** Most genes upregulated at meiotic initiation are bound by STRA8. (**A**) Overlap between the genes upregulated at meiotic initiation (see *Figure 1E*) and the STRA8-bound genes (see *Figure 2E*). The two gene lists significantly overlap, with 1,351 'STRA8-activated genes' that are directly bound and upregulated at meiotic initiation by STRA8 ($p<6.8\times10^{-322}$; one-tailed hypergeometric test). (**B**) Overlap between genes downregulated at meiotic initiation (see *Figure 1E*) and STRA8-bound genes. Downregulated genes are significantly depleted for STRA8-bound genes ($p<7.5\times10^{-99}$; one-tailed hypergeometric test). (**C**) Volcano plots, for STRA8-bound genes and all other genes, representing gene expression changes at meiotic initiation. Only expressed genes are shown. (**D**) Overlap between expressed STRA8-bound genes and genes differentially expressed at meiotic initiation. STRA8-bound genes are primarily upregulated; $p<2.2\times10^{-16}$, Fisher's exact test. (**E**) Gene expression fold changes at meiotic initiation. Boxplots show sample medians and interquartile ranges (IQRs), with whiskers extending no more than $1.5 \times$ IQR and outliers suppressed. ***$p<2.2\times10^{-16}$, one-tailed Mann-Whitney *U* test. (**F**) Volcano plot showing expression changes of 'STRA8-activated genes' in *Stra8*$^{+/-}$ preleptotene cells with high or low STRA8 levels. See *Figure 3—source data 1*.

DOI: https://doi.org/10.7554/eLife.43738.012

The following source data and figure supplement are available for figure 3:

**Source data 1.** Source data for RNA-seq and ChIP-seq analyses.

DOI: https://doi.org/10.7554/eLife.43738.014

**Figure supplement 1.** The breadth of the gene expression program directly upregulated at meiotic initiation by STRA8.

DOI: https://doi.org/10.7554/eLife.43738.013

chromatin regions and is unlikely to be a pioneer factor that initiates transcription of unexpressed genes. Rather, STRA8 functions to amplify transcriptional levels. This finding echoes our earlier analyses (*Figure 1F*), which revealed that upregulation at meiotic initiation primarily involves amplification of gene expression.

We then tested whether gene expression is upregulated in a sequence-specific manner. We searched for de novo motifs at the peaks of ChIP-seq read density near STRA8-activated promoters. The most enriched motif was concentrated at such peaks ($p<1\times10^{-140}$; *Figure 4B* and *Figure 4— figure supplement 1A*), and a similar sequence was the most enriched motif in the 400 bp window surrounding the TSS of STRA8-activated genes (*Figure 4—figure supplement 1B*). This motif features the core sequence CNCCTCAG. [Although the motif most closely resembles that of AP-2 family transcription factors (*Eckert et al., 2005*), the consensus AP-2 motif is not enriched at STRA8

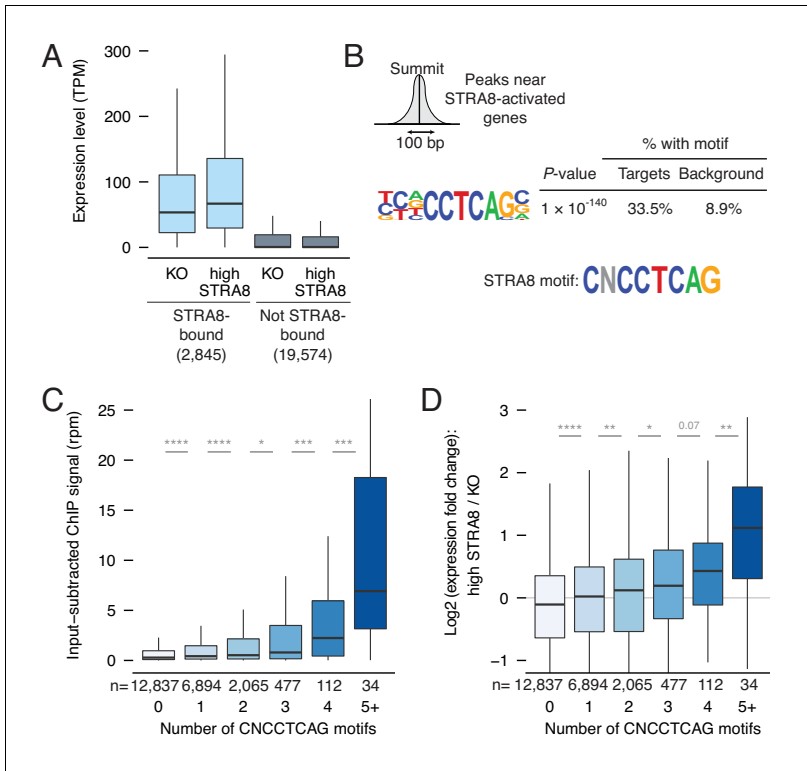

**Figure 4.** Mechanisms of gene activation at meiotic initiation. (**A**) Expression levels of genes in *Stra8* KO and high-STRA8 preleptotene cells. Genes are binned by whether they are STRA8-bound or not. (**B**) Identification of potential STRA8 binding sequences by HOMER de novo motif analysis. We searched the STRA8 binding peaks within promoters of STRA8-activated genes. Shown is a graphical depiction of the input to the motif-finding algorithm: the 100 bp windows surrounding the summits of ChIP-seq peaks. The top enriched motif has the consensus sequence CNCCTCAG. See *Figure 4—figure supplement 1* for other enriched motifs. (**C**) STRA8 ChIP-seq signal at gene promoters, with genes grouped by the number of promoter CNCCTCAG (perfect match) motifs. Boxplots show sample medians and interquartile ranges (IQRs), with whiskers extending no more than $1.5 \times$ IQR; outliers are suppressed. *p<0.01, , ***p<$1\times10^{-4}$, ****p<$1\times10^{-5}$, two-tailed Mann-Whitney *U* tests. (**D**) The expression fold change of genes at meiotic initiation, with genes grouped by number of promoter CNCCTCAG (perfect match) motifs. *p<0.01, **p<0.001, ****p<$1\times10^{-5}$, two-tailed Mann-Whitney *U* tests. See *Figure 4—source data 1*.

DOI: https://doi.org/10.7554/eLife.43738.015

The following source data and figure supplements are available for figure 4:

**Source data 1.** Source data for *Figure 4* panels.
DOI: https://doi.org/10.7554/eLife.43738.018
**Figure supplement 1.** Identification of a binding motif for STRA8-mediated upregulation at meiotic initiation.
DOI: https://doi.org/10.7554/eLife.43738.016
**Figure supplement 2.** STRA8 binding is not associated with AP-2 transcription factor motifs.
DOI: https://doi.org/10.7554/eLife.43738.017

binding sites, and AP-2 factors are not highly expressed in preleptotene cells (*Figure 4—figure supplement 2A–C*).]

If the CNCCTCAG motif mediates activation by STRA8 at meiotic initiation, then it should satisfy several predictions. First, motif density within the genome should correlate with STRA8 binding levels. This prediction was met: the motif count at ChIP-seq peak 'summits' correlated with the peak score (Spearman's ρ = 0.23, p<$1\times10^{-6}$ by permutation; *Figure 4—figure supplement 1C*). We observed as many as eight CNCCTCAG motifs in the 100 bp region surrounding a peak summit. Second, looking gene by gene, motif numbers should correlate with ChIP-seq binding levels. This prediction was also met: each additional motif at the promoter, from zero to five motifs total, was associated with increased STRA8 binding (*Figure 4C*). Third, genes with more motifs should be

more highly upregulated at meiotic initiation. This third prediction was met: greater numbers of CNCCTCAG motifs were associated with greater fold-changes in expression at meiotic initiation (*Figure 4D*). Altogether, these observations provide strong evidence that binding of STRA8 to the CNCCTCAG motif leads to upregulation of gene expression.

We note that upregulation at meiotic initiation is associated with CpG islands (CGI), stretches of DNA with high CpG dinucleotide frequency. 97% of STRA8-bound genes have CGI promoters, and the effect of the CNCCTCAG motif on transcriptional upregulation is accentuated in the vicinity of a CGI (*Figure 4—figure supplement 1D–F*).

## Amplification of a broad gene expression program at meiotic initiation

To understand the biological ramifications of the transcriptional changes at meiotic initiation, we explored the functions of the upregulated genes. We performed Gene Ontology (GO) analysis to identify cellular processes that were overrepresented among the 1,351 STRA8-activated genes, relative to all other expressed genes. Given our focus on meiotic initiation, we expected that meiosis-specific processes would be the most enriched. Meiosis-related categories (e.g. 'meiotic nuclear division,' 'meiosis I,' and 'meiotic cell cycle process') were indeed among the top categories, but the most significantly enriched category was 'cell cycle process' (2.2-fold enrichment, $p < 2.7 \times 10^{-17}$, one-tailed binomial test with Bonferroni correction; *Figure 5A*).

This led us to investigate whether genes upregulated at meiotic initiation have functions outside of germ cells. We asked whether STRA8-activated genes are expressed predominantly in the testis, because testis-specific gene expression in the adult can serve as a proxy for germ cell-specific function. Using a published RNA-seq dataset spanning nine adult male mouse tissues (eight somatic tissues plus testis) (*Merkin et al., 2012*), we considered a gene to be 'testis-biased' if its testis expression constituted more than half of its total expression summed across all nine tissues. As expected, the majority of catalogued meiotic prophase genes (*Soh et al., 2015*) are testis-biased (70%; *Figure 5B*). In contrast, only 16% of STRA8-activated genes are testis-biased, a percentage similar to that of all preleptotene-expressed genes, of which 12% are testis-biased (*Figure 5B*). Thus, this factor plays a key role in amplifying a program that includes but is not limited to meiosis genes.

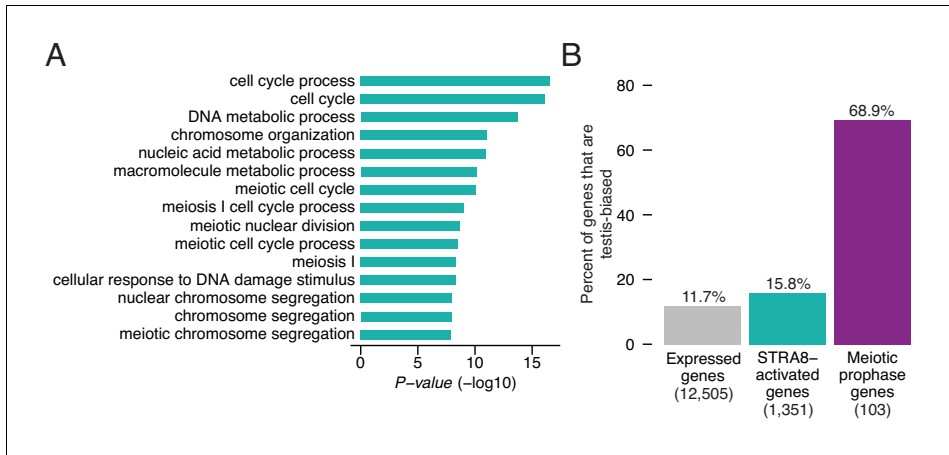

**Figure 5.** Upregulation of a broad gene expression program at meiotic initiation by STRA8. (**A**) The top 15 enriched Gene Ontology biological processes among the 1,351 STRA8-activated genes, compared to all preleptotene-expressed genes. *P*-values are from binomial tests with Bonferroni correction. (**B**) Fraction of genes that are testis-biased. See *Figure 5—source data 1*.
DOI: https://doi.org/10.7554/eLife.43738.019

The following source data is available for figure 5:

**Source data 1.** Source data for Figure 5 analyses.
DOI: https://doi.org/10.7554/eLife.43738.020

## Transcriptional amplification of meiotic prophase factors at meiotic initiation

Our analyses suggested that a common factor upregulates the program orchestrating meiotic chromosomal events. Although the *Stra8* gene is known to be required for robust expression of canonical meiotic genes (*Anderson et al., 2008*; *Baltus et al., 2006*; *Soh et al., 2015*), it was previously unknown whether the STRA8 protein directly activates their expression. To address this question, we compared STRA8 binding data against a published list of meiotic prophase genes, comprising 103 protein coding genes upregulated during meiotic prophase I in fetal ovarian germ cells (*Soh et al., 2015*). We find that, in males, 76 of these 103 genes are STRA8-bound, and 68 are STRA8-activated ($p<1.2\times10^{-28}$ and $p<8.2\times10^{-38}$, respectively, one-tailed hypergeometric test; *Figure 6A*; *Supplementary file 3*). STRA8 therefore directly upregulates this transcriptional program. Although the 68 STRA8-activated factors were robustly amplified, with a median fold-change of 3.8, 66 were

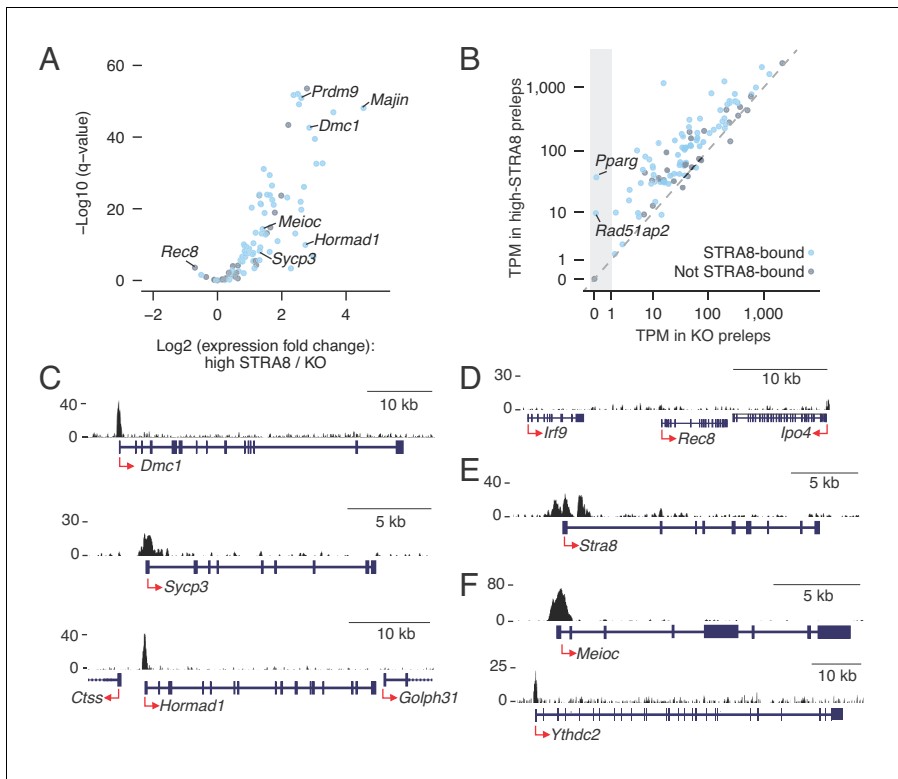

**Figure 6.** Coordinated upregulation of the meiotic prophase I gene expression program by STRA8. (**A**) Volcano plot depicting gene expression differences between high-STRA8 and *Stra8* KO preleptotene cells. Shown are 103 genes associated with meiotic prophase I in the fetal ovary (*Soh et al., 2015*), with 76 STRA8-bound genes shaded light blue. (**B**) Comparison of expression levels of meiotic prophase genes in high-STRA8 and KO preleptotene cells. The light gray region identifies genes that are not expressed in the KO. (**C**) Input-subtracted STRA8 ChIP-seq signal at promoters of key meiotic genes. Sequencing reads were pooled from three *Stra8*<sup>FLAG/FLAG</sup> ChIP replicates. Red arrows mark the TSS. (**D**) Lack of STRA8 ChIP-seq signal at *Rec8* gene. (**E**) STRA8 ChIP-seq signal at its own promoter. (**F**) STRA8 ChIP-seq signal at promoters of *Meioc* and *Ythdc2*..
DOI: https://doi.org/10.7554/eLife.43738.021

The following source data and figure supplements are available for figure 6:

**Source data 1.** Source data for CNCCTCAG motif enrichment at meiotic genes.
DOI: https://doi.org/10.7554/eLife.43738.024
**Figure supplement 1.** *Taf7l2* (*4933416C03Rik*) is a functional retrogene of *Taf7l*.
DOI: https://doi.org/10.7554/eLife.43738.022
**Figure supplement 2.** Enrichment and conservation of STRA8 (CNCCTCAG) motifs at promoters of meiotic genes.
DOI: https://doi.org/10.7554/eLife.43738.023

expressed even in *Stra8*-deficient preleptotene cells (*Figure 6B*). These results are consistent with our earlier finding that nearly all upregulated genes are expressed even in the absence of meiotic initiation (*Figure 1F*), and with published reports of meiotic gene expression even in mitotically dividing germ cells (*Evans et al., 2014*; *Wang et al., 2001*).

Previous data from fetal ovaries suggested a regulatory logic for the meiotic transcriptional program, with upregulation of most meiotic prophase genes being 'Stra8-dependent,' but a select few being completely 'Stra8-independent' (*Soh et al., 2015*). This model predicted STRA8's direct upregulation of *Stra8*-dependent genes (e.g. *Dmc1*, *Sycp3*, and *Hormad1*) but not of *Stra8*-independent genes (e.g. *Rec8*). We tested these predictions with our STRA8 ChIP-seq data, in the context of male meiosis. All nine genes that had been validated, by single-cell experiments, as *Stra8*-dependent in female meiosis (*Soh et al., 2015*) were STRA8-bound in male meiosis, and eight were significantly upregulated at male meiotic initiation. In fact, *Dmc1* was directly upregulated 7.3-fold, to 309 TPM (*Figure 6C*; *Supplementary file 3*). (The ninth gene, *Sycp1*, was upregulated by 1.4-fold at FDR q-value = 0.056.) In contrast, the *Stra8*-independent gene *Rec8* (*Koubova et al., 2014*; *Soh et al., 2015*) did not have a STRA8 ChIP-seq peak anywhere within 10 kb of its TSS (*Figure 6D*) and was significantly downregulated (1.6-fold) at meiotic initiation, as was observed in the embryonic ovary (*Soh et al., 2015*). *Rec8* expression thus appears to be completely independent of STRA8. Together, our ChIP-seq and RNA-seq data corroborate the nuanced control of the meiotic prophase transcriptional program, and suggest a similar regulatory logic in female (*Soh et al., 2015*) and male meiotic transcriptional programs: most but not all of the program is directly amplified by STRA8. We also found that STRA8 directly bound its own promoter (*Figure 6E*), suggesting a positive feedback loop in upregulation of this gene expression program.

STRA8 directly upregulates meiotic prophase genes with diverse functions, including meiotic cohesins (*Smc1b*, *Stag3*), synaptonemal complex proteins (*Syce1*, *Syce2*, *Sycp2*, *Sycp3*, *4930447C04Rik/Six6os1*), and meiotic telomere complex proteins (*Majin*, *Terb1*) (*Gómez-H et al., 2016*; *Handel and Schimenti, 2010*; *Shibuya et al., 2015*) (*Supplementary files 1* and *3*). *Spo11*, whose product catalyzes programmed meiotic double-strand breaks (DSBs) (*Keeney et al., 1997*), was STRA8-bound though not significantly upregulated in preleptotene cells. However, STRA8 upregulates *Top6bl* (*Gm960*), which encodes a protein that complexes with SPO11 (*Robert et al., 2016*), as well as *Prdm9*, which determines DSB sites (3.9- and 6.1-fold upregulation, respectively) (*Baudat et al., 2010*). Many other key meiotic recombination genes are significantly upregulated by STRA8: *Mei1*, *Mei4*, *Hormad1*, *Hormad2*, *Dmc1*, *Rad51*, *Brca2*, *Msh5*, *Tex11*, and *Mlh1* (*Baudat et al., 2013*). Furthermore, STRA8 directly upregulates the expression of *Meioc* and *Ythdc2* (2.6- and 2.0-fold, respectively), which encode post-transcriptional regulators without which germ cells initiate meiosis but undergo a premature and abnormal metaphase before eventually undergoing apoptosis (*Figure 6F*) (*Abby et al., 2016*; *Bailey et al., 2017*; *Jain et al., 2018*; *Soh et al., 2017*). In addition, the meiosis-specific cyclin *Ccnb3*, which is expressed early in meiotic prophase I and prevents premature progression through the cell cycle (*Nguyen et al., 2002*), was STRA8-bound in one ChIP-seq replicate and was 6.9-fold upregulated at meiotic initiation (*Supplementary files 1* and *3*). STRA8's direct upregulation of all such genes explains why *Stra8*-deficient germ cells fail to initiate meiosis.

Some transcriptional regulators required for proper meiosis are also upregulated at meiotic initiation. One such STRA8-activated gene encodes the testis-enriched transcriptional regulator MYBL1, without which males do not successfully complete meiosis (*Bolcun-Filas et al., 2011*). STRA8 amplifies the expression of *Taf7l* and *Taf4b*, which encode germ cell-specific or -enriched components of the TFIID complex that are needed for proper gametogenesis (*Cheng et al., 2007*; *Falender et al., 2005*; *Grive et al., 2016*; *Zhou et al., 2013*). STRA8 also directly upregulates *4933416C03Rik* (now called *Taf7l2*), which appears to be a functional rodent-specific retrogene of *Taf7l*. The *Taf7l2* gene is robustly expressed in preleptotene cells (~70–115 TPM) and is highly testis-biased, with its testis expression comprising >99% of its total expression across different tissues (*Figure 6—figure supplement 1A–C*; *Supplementary file 2*). Additionally, STRA8 activates the transcription factor *Pparg*, whose function has been extensively studied in adipose tissue; it is not germ cell specific. Although *Pparg* was previously shown to be upregulated during meiotic prophase (*Chen et al., 2018*; *Soh et al., 2015*), its role in meiosis has not been probed. STRA8 strongly binds the *Pparg* promoter, which has eight STRA8 (CNCCTCAG) motifs, and activates its expression from 'off' (TPM < 0.01) in *Stra8*-deficient cells to robust expression (TPM of 38) in high-STRA8 preleptotene

cells (*Figure 6B*). Thus, PPARG is a candidate to be an important transcriptional regulator during meiosis. STRA8's upregulation of this and other transcriptional regulators may extend and reinforce the transcriptional programs on which meiosis depends.

Finally, we observed that, at meiotic initiation, STRA8 binds and upregulates the germ cell marker genes *Dazl* and *Gcna* (1.7- and 3.3-fold, respectively). (Note: *Gcna* was STRA8-bound in only one of three ChIP-seq replicates; *Supplementary file 2*). Both are robustly expressed in male and female germ cells from when they arrive at the gonad (*Carmell et al., 2016*; *Hu et al., 2015*), with *Dazl* functioning upstream of *Stra8* to provide competency to enter meiosis (*Gill et al., 2011*; *Lin et al., 2008*). Their STRA8-mediated upregulation hints at ongoing functions during meiosis.

Upregulation of meiotic genes likely depends on STRA8 binding to the CNCCTCAG motif. We observed that most meiotic prophase genes have at least one CNCCTCAG motif at the promoter, and that meiotic gene promoters are significantly enriched for CNCCTCAG motifs compared to all other genes ($p<3.5\times10^{-16}$, two-tailed Mann Whitney $U$ test; *Figure 6—figure supplement 2A,B*). Several meiotic genes had more than six STRA8 binding motifs at their promoters, including *4930447C04Rik/Six6os1* (14 motifs), *Pparg* (8), *M1ap* (7), *Prdm9* (6), *Syce3* (6), and *4933416C03Rik/Taf7l2* (6) (*Supplementary file 3*). Furthermore, human orthologs of meiotic genes were also enriched for CNCCTCAG motifs ($p<1.1\times10^{-8}$, two-tailed Mann-Whitney $U$ test; *Figure 6—figure supplement 2A,B*), suggesting that upregulation of meiotic genes by STRA8 binding to the CNCCTCAG motif is conserved in humans.

## Upregulation of cell cycle-associated genes at meiotic initiation

The most enriched GO category in the ensemble of genes upregulated at meiotic initiation by STRA8 was 'cell cycle,' raising the possibility that STRA8 also amplifies the expression of general cell-cycle genes that function in both mitotic and meiotic cell cycles. To exclude the possibility that this enrichment was simply due to an enrichment of canonical meiotic prophase genes within the broader 'cell cycle' category, we tested this association in two ways. First, we asked whether STRA8-activated genes are enriched for cell-cycle genes after excluding genes associated with the term 'meiotic cell cycle'. Indeed, non-meiotic cell-cycle genes were still overrepresented among STRA8-activated genes ($p<5.0\times10^{-6}$, one-tailed binomial test with Bonferroni correction; *Figure 7A*). Next, we compared the STRA8-activated genes with an independently curated list of ~200 genes associated with cellular division, which includes genes extensively studied in non-meiotic contexts that function in DNA replication, cell-cycle regulation, DNA damage, cytokinesis, or at the kinetochore (*McKinley and Cheeseman, 2017*). STRA8-activated genes were significantly enriched in this curated list ($p<9.1\times10^{-9}$, one-tailed hypergeometric test; *Figure 7A*), confirming STRA8's direct role in upregulating a cell cycle-associated program.

Motivated by previous findings that *Stra8* is genetically required for meiotic S phase (*Baltus et al., 2006*; *Dokshin et al., 2013*), we asked whether STRA8 directly regulates the G1-S transition of the cell cycle. We found evidence for its upregulation of G1-S regulators and DNA-replication genes. STRA8 bound and amplified by 3.0-fold the expression of *E2f1*, a key transcription factor that induces G1-S gene expression (*Figure 7B,C*) (*Bracken et al., 2004*). STRA8 binding also was associated with significant upregulation of *Ccne1* and *Ccne2* (*Figure 7B,C* and *Supplementary file 2*), which encode cyclins that regulate the cyclin dependent kinase CDK2 at the G1-S transition (*Caldon and Musgrove, 2010*); *Cdk2* itself is also STRA8-activated at meiotic initiation (*Figure 7D*). Furthermore, STRA8 upregulates components of the pre-replicative complex for DNA replication: members of the origin recognition complex (ORC4 and ORC6), subunits of the replicative helicase (MCM4, MCM5, and MCM6), and CDC6. Genes that encode DBF4 and CDC7, which together stimulate the MCM complex to initiate DNA replication, are also upregulated by STRA8 (*Figure 7D* and *Supplementary file 2*).

These observations led us to test whether STRA8 regulates a broader cell cycle-associated transcriptional program. Specifically, we asked whether STRA8-activated genes also are targets of the transcription factors E2F1 and E2F4, which drive a broad gene expression program for dividing cells, including genes required for DNA replication, cell-cycle control, DNA damage repair, and chromosome segregation (*Bracken et al., 2004*); or targets of the FOXM1 transcription factor, which regulates many genes required for the G1-S and G2-M transitions (*Laoukili et al., 2005*; *Wang et al., 2005*). We identified targets of these factors by analyzing ENCODE consortium ChIP-seq data from various non-meiotic cell lines. We found that more than half of STRA8-activated genes were also

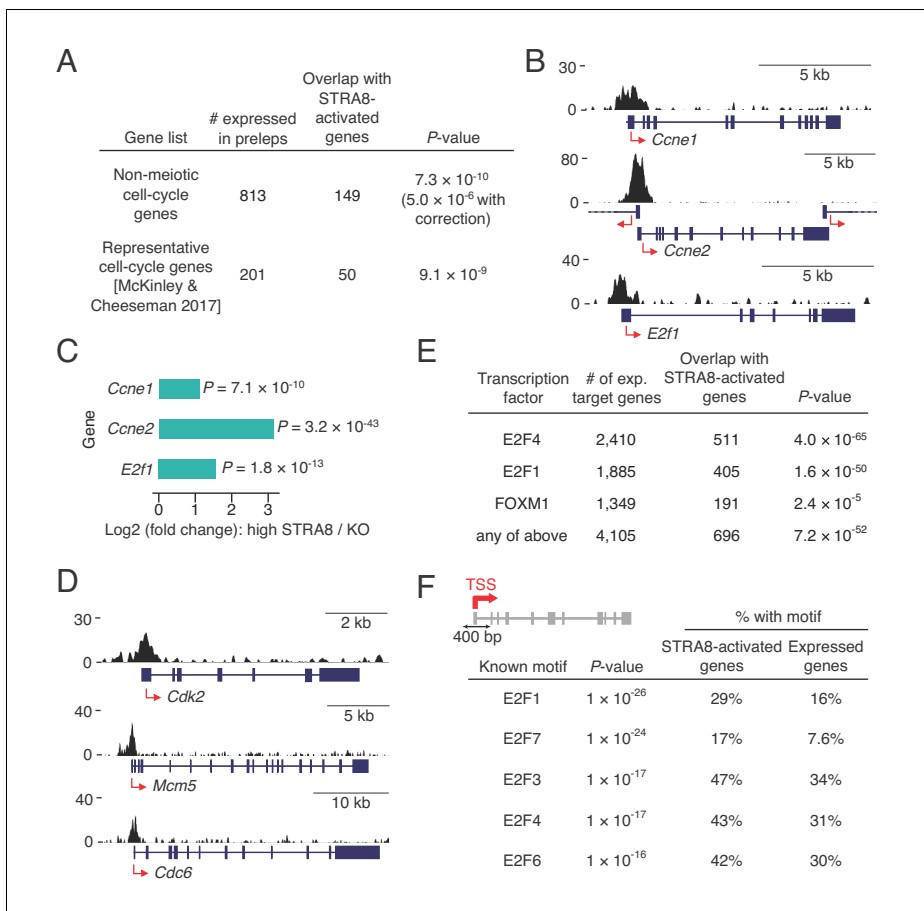

**Figure 7.** Coordinated upregulation of a transcriptional program for cell-cycle progression at meiotic initiation. (A) Overlap of STRA8-activated genes at meiotic initiation with cell-cycle associated gene lists. *P*-value for the non-meiotic cell cycle GO terms was calculated using a binomial test with Bonferroni correction. *P*-value for overlap with a list of representative cell-cycle genes was calculated using a one-tailed hypergeometric test. (B) Input-subtracted STRA8 ChIP signal showing STRA8 binding at promoters of key G1-S transition genes. Sequencing reads were pooled from the three *Stra8*[FLAG/FLAG] ChIP replicates. (C) Expression changes of key G1-S genes at meiotic initiation. (D) *Stra8*[FLAG/FLAG] ChIP-seq signal at promoters of other key cell-cycle genes. (E) Overlap of STRA8-activated genes with targets of known cell cycle-driving transcription factors. E2F4-, E2F1-, and FOXM1-bound genes were identified from ENCODE consortium ChIP-seq datasets. *P*-values were obtained by one-tailed hypergeometric tests. (F) Top five known motifs enriched near the TSS of STRA8-activated genes, compared to all other expressed genes. Enrichment of E2F transcription factor motifs suggests STRA8's role in regulating a cell cycle-associated transcriptional program. Note that the E2F6 motif enrichment could also reflect the known role of E2F6, a non-canonical E2F factor, in regulating the expression of meiotic genes (*Kehoe et al., 2008*; *Pohlers et al., 2005*).
DOI: https://doi.org/10.7554/eLife.43738.025

targets of one or more of these other cell cycle transcription factors (*Figure 7E*). Furthermore, at the TSS of STRA8-activated genes, the most enriched known motifs were those for E2F transcription factors, which suggests that the genes are regulated in a cell cycle-dependent manner (*Figure 7F*). Together, our findings demonstrate that a broad cell cycle-associated transcriptional program is amplified by STRA8 at the onset of meiotic initiation.

## Discussion

Prior to the studies reported here, how germ cells in multicellular organisms initiate meiosis was not well understood. By specifically examining mouse germ cells entering meiosis in vivo (*Romer et al., 2018*), we now show that meiotic initiation entails the robust upregulation of a broad gene

ensemble in the preleptotene stage. We demonstrate that STRA8, a transcriptional activator, triggers the unique meiotic cell cycle by amplifying the expression of meiotic prophase I genes, G1-S cell-cycle genes, and factors that specifically inhibit the mitotic program.

We were able to explore the switch to a meiotic cell cycle in unprecedented detail because we could measure transcriptional changes directly within the preleptotene population before and after meiotic initiation. We found that most genes upregulated at meiotic initiation, including canonical meiotic prophase genes, are also expressed in preleptotene cells in the absence of meiotic initiation, providing context for earlier, anecdotal reports of meiotic gene expression in pre-meiotic germ cells (*Evans et al., 2014*; *Jan et al., 2017*; *Wang et al., 2001*). Expression of these genes alone is therefore not enough to trigger meiotic initiation, and should not be taken as evidence of successful meiotic entry (*Handel et al., 2014*). Rather, what triggers meiotic initiation may be the coordinated upregulation of most canonical meiotic prophase I genes during the preleptotene stage. A common factor, STRA8, likely upregulates their expression past threshold levels required to support the events of homolog pairing, synapsis, and meiotic recombination (*Figure 8A*). We find, for example, that STRA8 upregulates key meiotic genes *Prdm9*, *Dmc1*, and *Hormad1* more than 4-fold, and the meiotic telomere complex factor *Majin* by 23-fold. Also critical for triggering meiosis may be the small subset of STRA8-activated genes that are initially 'off' in the absence of meiotic initiation, such as *Rad51ap2* and *Pparg*, whose meiotic functions remain unexplored. In addition, meriting further investigation are STRA8-activated, testis-biased genes of unknown function (*Supplementary file 1*).

Our work highlights STRA8 as a key transcriptional regulator for initiation of meiosis. Although its CNCCTCAG binding motif differs from the typical bHLH binding motif CANNTG (*Massari and Murre, 2000*), our results are consistent with STRA8 functioning as a bHLH transcription factor. We find that a STRA8 isoform lacking the bHLH domain (*Baltus et al., 2006*; *Tedesco et al., 2009*) cannot initiate meiosis or localize to the nucleus in vivo, paralleling a prior in vitro finding that its bHLH domain directs nuclear localization (see Appendix 1) (*Tedesco et al., 2009*). Because HLH domains mediate dimerization, we propose that STRA8's HLH domain heterodimerizes with that of another HLH factor before it translocates to the nucleus to bind gene promoters. STRA8's promoter specificity stands out; few transcription factors exhibit such a preponderance (84%) of binding at promoters (*Neph et al., 2012*). Perhaps it regulates transcription by releasing promoter-paused RNA polymerase II (RNAPII) into productive elongation, which in many developmental contexts facilitates rapid and synchronous gene upregulation (*Adelman and Lis, 2012*). Alternatively, STRA8 may facilitate

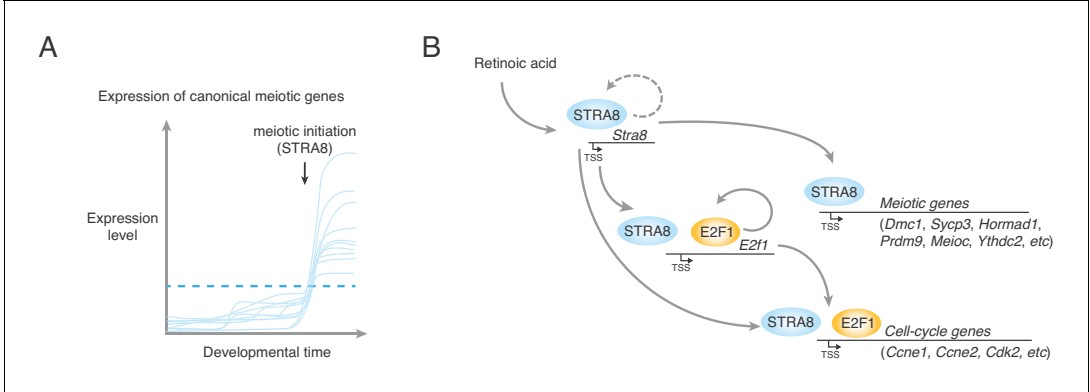

**Figure 8.** A model for upregulation of the transcriptional program that triggers the meiotic cell cycle. (**A**) Upregulation of meiotic prophase genes at meiotic initiation. Shown are hypothetical meiotic genes and their expression levels across developmental time. Many meiotic genes are expressed at detectable levels before meiotic initiation. We propose that their upregulation above a threshold level (dotted blue line) required for events such as synapsis and meiotic recombination triggers the execution of the chromosomal events of meiotic prophase I. In preleptotene spermatocytes, STRA8 directly upregulates this ensemble of genes. (**B**) A model of STRA8-mediated gene expression changes at meiotic initiation. First, retinoic acid induces expression of *Stra8*. STRA8 then activates a broad transcriptional program by directly binding promoters of both meiotic genes and cell-cycle genes. Among STRA8-activated cell cycle genes is *E2f1*, which is known to drive the G1/S transition. E2F1 upregulates its own expression as well as a broad ensemble of cell cycle-associated genes, including those involved in a positive feedback loop to reinforce cell-cycle entry. STRA8 also binds its own promoter, potentially establishing a feedback loop that further commits germ cells to the meiotic cell cycle.
DOI: https://doi.org/10.7554/eLife.43738.026

transcription by recruiting RNAPII, the pre-initiation complex, or chromatin modifiers such as histone acetyltransferases. STRA8 is unlikely to be a pioneer transcription factor, because its preference for promoters of already-expressed genes indicates that it primarily binds open chromatin regions.

We suggest that meiotic initiation involves the amplification of a broad gene regulatory network that drives not only the chromosomal events of meiotic prophase I but also cell-cycle entry (*Figure 8B*). In mice, STRA8 activates DNA-replication and G1-S genes, corroborating and extending previous genetic evidence that meiotic initiation takes place upstream of meiotic S phase (*Baltus et al., 2006*; *Dokshin et al., 2013*). STRA8 directly activates *E2f1*, which drives the G1-S cell-cycle transition (*Bracken et al., 2004*). In primary human fibroblasts, activation of the E2F1-CCNE1/2-CDK2 positive feedback loop signals passage through the 'restriction point' (*Schwarz et al., 2018*), a point-of-no-return indicating commitment to the cell-cycle program (*Bertoli et al., 2013*). STRA8's coordinated upregulation of these feedback-loop genes suggests that initiation of the meiotic program (encompassing meiotic DNA replication to the meiotic divisions) involves passage through a meiotic version of the 'restriction point.' STRA8 also binds its own promoter, suggesting potential positive feedback to promote robust upregulation of meiotic S-phase and prophase genes, together with cell-cycle transcription factors such as *E2f1*. As in mitotic G1-S, positive feedback during meiotic G1-S may be followed by negative feedback (*Bertoli et al., 2013*): fetal ovarian germ cells downregulate *Stra8* after meiotic entry (*Soh et al., 2015*), and STRA8 levels in male germ cells decline precipitously after the leptotene stage (*Zhou et al., 2008*).

The upregulation of general cell-cycle genes begs the question, how does the preleptotene cell know to induce a meiotic (and not a mitotic) program? Specificity for meiosis may result in part from upregulation of genes that ensure a meiosis-specific cell-cycle state. First, STRA8 directly upregulates the genes encoding the post-transcriptional regulators MEIOC and YTHDC2, which are critical for the unique meiotic cell cycle state; they prevent aberrant expression of mitotic genes during meiotic prophase I and help meiotic germ cells maintain the extended prophase unique to meiosis I (*Abby et al., 2016*; *Bailey et al., 2017*; *Jain et al., 2018*; *Soh et al., 2017*). [Note that our RNA-seq analyses provide no insight into post-transcriptional regulation of meiosis, which is known to be pervasive in yeast meiosis (*Brar et al., 2012*).] Second, we find that *Ccnb3*, a meiosis-specific cyclin that prevents precocious progression through meiotic prophase (*Nguyen et al., 2002*), is 6.9-fold upregulated at meiotic initiation. Finally, the STRA8-regulated genes *Ccne2* and *Cdk2*, which are considered 'general' cell-cycle genes, may have meiosis-specific roles. These genes are not strictly required for mitotic proliferation but are necessary for proper homolog pairing and synapsis in meiosis (*Martinerie et al., 2014*; *Ortega et al., 2003*).

The far-reaching role of a common factor – STRA8 – in coordinating broad transcriptional changes raises the possibility of whether it alone is sufficient to initiate the meiotic program. We would argue that it is not sufficient, and that triggering meiosis instead entails amplification of a transcriptional program that exists only in the early preleptotene stage. In fact, STRA8 is expressed in, but does not induce meiosis in, early (type A) spermatogonia (*Zhou et al., 2008*). Furthermore, ectopic STRA8 expression in intermediate type spermatogonia (*Endo et al., 2015*) (only two cell divisions prior to the preleptotene stage) or in vitro-derived germ cells (*Miyauchi et al., 2017*) is not alone sufficient to trigger meiosis. Competency for meiotic initiation may thus require a chromatin or transcriptional state established in preceding mitotic divisions (*Zhang et al., 2014*), including the observed lower-level expression of meiotic genes (*Evans et al., 2014*; *Soh et al., 2015*; *Wang et al., 2001*). We suggest that this preleptotene-specific competency intersects with STRA8's function as a transcriptional activator to amplify and orchestrate the unique gene regulatory network that drives meiosis. In other multicellular organisms, transcriptional regulators of meiotic initiation remain to be identified (*Kimble, 2011*). However, the combination of pre-existing transcriptional programs together with a dramatic amplification of a broad meiosis-associated gene ensemble may be a common strategy to trigger the one meiotic cell cycle between successive generations with precision, only at the proper time and place.

## Materials and methods

**Key resources table**

*Continued on next page*

*Continued*

| Reagent type (species) or resource | Designation | Source or reference | Identifiers | Additional information |
|---|---|---|---|---|
| Gene (*Mus musculus*) | *Stimulated by retinoic acid 8 (Stra8)* | Mouse Genome Informatics | MGI:107917 | |
| Strain, strain background (*M. musculus*) | B6 | Taconic | TAC: B6-F | C57BL/6NTac |
| Strain, strain background (*M. musculus*) | B6/129 | Taconic | TAC: B6129-F | B6129F1/Tac |
| Strain, strain background (*M. musculus*) | CD-1 | Charles River Laboratories | CRL: 022 | Crl:CD1(ICR) |
| Genetic reagent (*M. musculus*) | *Stra8$^{FLAG}$* | this paper | | FLAG-tagged knock-in allele generated by CRISPR/Cas9 |
| Genetic reagent (*M. musculus*) | *Stra8$^{\Delta121}$* | this paper | | *Stra8* allele with a 121-bp deletion generated by CRISPR/Cas9 |
| Genetic reagent (*M. musculus*) | *Stra8*-deficient allele, *Stra8$^-$* | The Jackson Laboratory | JAX: 023805; MGI:3622304 | *Stra8$^{tm1Dcp}$* |
| Genetic reagent (*M. musculus*) | *Ddx4-Cre* | PMID:23858447 | MGI:5554579 | *Ddx4$^{tm1.1(cre/mOrange)Dcp'}$* |
| Genetic reagent (*M. musculus*) | *Rosa26-tdTomato* | The Jackson Laboratory | JAX: 007914 | B6.Cg-Gt(ROSA)26 Sor$^{tm14(CAG-tdTomato)Hze}$/J |
| Cell line (*M. musculus*) | v6.5 embryonic stem cells | other | RRID:CVCL_C865 | Jaenisch Lab, Whitehead Institute |
| Biological sample (*M. musculus*) | CF6Neo Mouse Embryonic Fibroblasts (MEF), Mitomycin C Treated | MTI-GlobalStem | GlobalStem:GSC-6105M | |
| Antibody | anti-STRA8 (rabbit polyclonal) | Abcam | Abcam:ab49405; RRID:AB_945677 | (IF 1:250; IHC 1:500: WB 1:1000) |
| Antibody | anti-STRA8 (rabbit polyclonal) | Abcam | Abcam:ab49602; RRID:AB_945678 | (IF 1:250; IHC 1:500: WB 1:1000) |
| Antibody | anti-FLAG M2 (mouse monoclonal) | MilliporeSigma | MilliporeSigma:F1804; RRID:AB_262044 | (IHC 1:200) |
| Antibody | anti-FLAG M2, HRP-conjugated (mouse monoclonal) | MilliporeSigma | MilliporeSigma:A8592; RRID: AB_439702 | (WB 1:1000) |
| Antibody | anti-DMC1 H100 (rabbit polyclonal) | Santa Cruz Biotechnology | SantaCruz:sc-22768; RRID:AB_2277191 | (IF 1:250) |
| Antibody | anti-SCP3 D-1 (mouse monoclonal) | Santa Cruz Biotechnology | SantaCruz:sc-74569; RRID:AB_2197353 | (IF 1:250) |
| Antibody | anti-phospho-histone H2AX (Ser139), clone JBW301 (mouse monoclonal) | MilliporeSigma | MilliporeSigma:05–636; RRID:AB_309864 | (IF1:250) |
| Antibody | anti-DDX4/MVH (goat polyclonal) | R and D Systems | R and D Systems: AF2030; RRID:AB_2277369 | (IF 1:500) |

*Continued on next page*

*Continued*

| Reagent type (species) or resource | Designation | Source or reference | Identifiers | Additional information |
|---|---|---|---|---|
| Antibody | anti-alpha tubulin, HRP-conjugated (rabbit polyclonal) | Abcam | Abcam:ab40742; RRID:AB_880625 | (WB 1:5000) |
| Antibody | anti-goat Alexa Flour 488 (donkey polyclonal) | Jackson ImmunoResearch | JacksonImmuno Research:705-546-147 | (IF 1:250) |
| Antibody | anti-rabbit Rhodamine Red-X (RRX) (donkey polyclonal) | Jackson ImmunoResearch | JacksonImmuno Research:711-295-152 | (IF 1:250) |
| Antibody | anti-mouse Cy5 (donkey polyclonal) | Jackson ImmunoResearch | JacksonImmuno Research:715-175-150 | (IF 1:250) |
| Antibody | anti-rabbit, peroxidase -conjugated (donkey polyclonal) | Jackson ImmunoResearch | JacksonImmuno Research:711-035-152 | (WB 1:5000) |
| Recombinant DNA reagent | pX330-U6-Chimeric_BB-CBh-hSpCas9 (plasmid) | Addgene | Addgene:42230 | |
| Sequence-based reagent | primers and oligonucleotides used in this study | this paper | | All primers and oligonucleotides used in this study are available in *Supplementary file 6* |
| Chemical compound, drug | N,N′-Octamethy lenebis(2,2-dichlo roacetamide) [Win18,446] | Santa Cruz Biotechnology | SantaCruz:sc-295819 | |
| Chemical compound, drug | Retinoic acid | MilliporeSigma | MilliporeSigma:R2625 | |
| Software, algorithm | FASTX Toolkit | other | RRID:SCR_005534 | http://hannonlab.cshl.edu/fastx_toolkit/ |
| Software, algorithm | Bowtie1 (v.1.2.0) | PMID:19261174 | RRID:SCR_005476 | http://bowtie-bio.sourceforge.net/index.shtml |
| Software, algorithm | Samtools (v1.5) | PMID:19505943 | RRID:SCR_002105 | http://samtools.sourceforge.net/ |
| Software, algorithm | MACS2 (v2.1.1.20160309) | PMID:18798982 | RRID:SCR_013291 | https://github.com/taoliu/MACS |
| Software, algorithm | htseq count (v0.8.0) | PMID:25260700 | RRID:SCR_011867 | https://htseq.readthedocs.io |
| Software, algorithm | kallisto (v0.43.0) | PMID:27043002 | RRID:SCR_016582 | https://pachterlab.github.io/kallisto/about |
| Software, algorithm | DESeq2 (v1.18.1) | PMID:25516281 | RRID:SCR_015687 | https://bioconductor.org/packages/release/bioc/html/DESeq2.html |
| Software, algorithm | PANTHER (v13.1) | PMID:27899595 | RRID:SCR_004869 | http://www.pantherdb.org/ |
| Software, algorithm | HOMER (v4.9.1) | PMID: 20513432 | RRID:SCR_010881 | http://biowhat.ucsd.edu/homer/index.html |
| Software, algorithm | deepTools (v2.5.3) | PMID:27079975 | RRID:SCR_016366 | http://deeptools.readthedocs.io/en/latest/index.html |
| Software, algorithm | RStudio v1.1.414 | RStudio | RRID:SCR_000432 | |

*Continued on next page*

Continued

| Reagent type (species) or resource | Designation | Source or reference | Identifiers | Additional information |
|---|---|---|---|---|
| Software, algorithm | PHYLIP (v3.66) | Other | RRID:SCR_006244 | Distributed by J Felsenstein. http://evolution.genetics .washington.edu /phylip.html |
| Commercial assay or kit | Periodic Acid-Schiff (PAS) Kit | MilliporeSigma | MilliporeSigma:395B | |
| Commercial assay or kit | Mouse on Mouse ImmPRESS HRP (Peroxidase) Polymer Kit | VECTOR Laboratories | VectorLabs:MP2400 | |
| Commercial assay or kit | ImmPRESS HRP Anti-Rabbit IgG (Peroxidase) Polymer Detection Kit, made in Horse | VECTOR Laboratories | VectorLabs:MP-7401 | |
| Commercial assay or kit | mMESSAGE mMACHINE T7 Kit | Thermo Fisher Scientific | Thermo Fisher:AM1344 | |
| Commercial assay or kit | MEGAsh ortscript T7 Kit | Thermo Fisher Scientific | Thermo Fisher:AM1354 | |
| Commercial assay or kit | MEGAclear Kit | Thermo Fisher Scientific | Thermo Fisher:AM1908 | |
| Commercial assay or kit | ChIP DNA Clean and Concentrator Kit | Zymo Research | Zymo:D5205 | |
| Commercial assay or kit | TruSeq ChIP Sample Preparation Kit | Illumina | Illumina:IP -202–1024 | |
| Commercial assay or kit | Accel-NGS 2S Plus DNA Library Kit | Swift Biosciences | SwiftBio:21024 | |
| Commercial assay or kit | TruSeq Stranded mRNA Library Prep Kit | Illumina | Illumina:20020594 | |
| Other | Hematoxylin | Life Technologies | LifeTech:008011 | |
| Other | 2.5% Normal Horse Serum Blocking Solution | VECTOR Laboratories | VectorLabs:S-2012 | |
| Other | ImmPACT DAB Peroxidase (HRP) Substrate | VECTOR Laboratories | VectorLabs:SK-4105 | |
| Other | Normal donkey serum | Jackson ImmunoResearch | JacksonImmunoResearch:017000121 | |
| Other | VECTASHIELD Antifade Mounting Media for Fluorescence | VECTOR Laboratories | VectorLabs:H-1000 | |
| Other | Collagenase, Type I | Worthington Biochemical | Worthington: LS004196 | |
| Other | Hyaluronidase | MilliporeSigma | MilliporeSigma:H3506 | |
| Other | DNaseI | MilliporeSigma | MilliporeSigma:D5025 | |
| Other | Benzonase Nuclease | MilliporeSigma | MilliporeSigma:70664–3 | |
| Other | Protease inhibitor, EDTA Free | MilliporeSigma | MilliporeSigma: 11836170001 | |
| Other | ESGRO Recombinant Mouse LIF Protein | MilliporeSigma | Millipore Sigma:ESG1107 | |
| Other | Dynabeads Protein G for Immunoprecipitation | Thermo Fisher Scientific | Thermo Fisher:10004D | |

*Continued on next page*

*Continued*

| Reagent type (species) or resource | Designation | Source or reference | Identifiers | Additional information |
|---|---|---|---|---|
| Other | Lumi-Light Western Blotting Substrate | MilliporeSigma | Millipore Sigma:12015200001 | |
| Other | DirectPCR Lysis Reagent (Mouse Tail) | Viagen Biotech | Viagen:102 T | |

## Mice

All mouse experiments were approved by the Massachusetts Institute of Technology (MIT) Division of Comparative Medicine, which is overseen by the MIT Institutional Animal Care and Use Committee (IACUC). For embryonic gonad collection, females were housed with males overnight, and 12:00 p.m. (noon) of the day the vaginal plug was observed was considered to be embryonic day 0.5. To generate $Stra8^{FLAG/FLAG}$ mice, we either mated $Stra8^{FLAG/FLAG}$ homozygous mice to each other or mated $Stra8^{+/FLAG}$ heterozygous mice to each other. To generate $Stra8^{\Delta121/\Delta121}$ mice, which do not express the long STRA8 isoform, $Stra8^{+/\Delta121}$ heterozygous mice were mated to each other. $Stra8^{FLAG}$ and $Stra8^{\Delta121}$ alleles used in phenotypic characterizations of $Stra8^{FLAG/FLAG}$ and $Stra8^{\Delta121/\Delta121}$ mice were backcrossed to C57BL/6NTac (B6) for at least seven generations (>99.5% B6); ChIP data were generated from mice that were >96% B6. For lineage sorting of germ cells, we used the $Ddx4^{Cre}$ allele (official allele name $Ddx4^{tm1.1(cre/mOrange)Dcp}$) (*Hu et al., 2013*), backcrossed at least nine generations to B6, and the $Rosa26^{tdTomato}$ allele (B6.Cg-$Gt(ROSA)26Sor^{tm14(CAG-tdTomato)Hze}$/J from the Jackson Laboratory, Bar Harbor, ME) (*Madisen et al., 2010*), backcrossed at least 10 generations to B6. The $Stra8^{tm1Dcp}$ allele (*Baltus et al., 2006*) was backcrossed to B6 at least 28 generations to generate $Stra8$-deficient ($Stra8^{-/-}$) mice. B6 and B6129F1/Tac (B6/129) mice were obtained from Taconic Biosciences (Rensselaer, NY), and Crl:CD1(ICR) (CD-1) mice were obtained from Charles River Laboratories (Wilmington, MA).

## Mouse genotyping

An ear biopsy (for postnatal mice) or a tail piece (from embryonic samples) was lysed in DirectPCR Lysis Reagent (Mouse Tail) (Viagen Biotech, Los Angeles, CA) supplemented with 400 µg/mL Proteinase K at 55°C overnight and then incubated at 85°C for 45 min. This crude lysate was used for genotyping by PCR with the primers listed in *Supplementary file 6*.

## Generation of $Stra8^{FLAG}$ and $Stra8^{\Delta121}$ alleles

Modified *Stra8* mouse alleles were generated by one-cell embryo injection of CRISPR/Cas9 reagents, following published protocols (*Wang et al., 2013*; *Yang et al., 2013*). Briefly, an oligonucleotide pair targeting exon 2 of *Stra8* was cloned into the guide RNA (gRNA) cassette of the vector pX330 (Addgene, Cambridge, MA), which also contains a Cas9 expression cassette (*Cong et al., 2013*). gRNA and Cas9 RNAs were transcribed in vitro using the MEGAshortscript T7 kit (Thermo Fisher, Waltham, MA) and mMESSAGE mMACHINE T7 ULTRA kit (Thermo Fisher), respectively, and purified using the MEGAclear kit (Thermo Fisher). RNAs were eluted in RNase free water. To obtain embryos for injection, superovulated female B6/129 mice (~9 weeks of age) were mated to B6 stud males. Fertilized one-cell embryos were injected cytoplasmically with a mixture of gRNA (50 ng/µL), Cas9 (50 ng/µL), and a 200 bp single-stranded oligo (100 ng/µL) for homology directed repair (HDR) following Cas9-mediated double-strand break (DSB) formation. This oligo, ordered as an Ultramer DNA oligo from Integrated DNA Technologies (Coralville, IA), encodes a FLAG peptide and an 8-aa linker sequence (GSGSGSGS), flanked on both sides by ~75 bp of sequence homologous to the *Stra8* locus. The injected embryos were cultured at 37°C under 5% $CO_2$ until 3.5 days post coitum (dpc), at which time they were at blastocyst stage. These blastocysts were transferred into the uteri of 2.5-dpc pseudopregnant CD-1 females. Genotyping of post-natal mice was performed by PCR amplification followed by sequencing, which confirmed the generation of the mutant *Stra8* alleles: $Stra8^{FLAG}$ (by successful repair of the Cas9-mediated DSB by HDR with the oligo) and $Stra8^{\Delta121}$ (by repair of the DSB with non-homologous end joining, resulting in a 121 bp deletion). See *Supplementary file 6* for a list of oligos and primers used.

## Histological analysis

Dissected tissues were fixed in Bouin's solution at room temperature (3 hr or overnight) or in 4% (wt/vol) paraformaldehyde at 4°C (overnight). Fixed tissues were embedded in paraffin and sectioned to 5 µm thickness. Slides were dewaxed in xylenes, rehydrated with an ethanol gradient, and heated in citrate buffer (10 mM sodium citrate, 0.05% Tween 20, pH 6.0) in the microwave.

For fluorescent detection, slides were first blocked with 5% donkey serum (Jackson ImmunoResearch, West Grove, PA) and then incubated with primary antibodies diluted in 5% donkey serum. Primary antibodies were diluted 1:500 for DDX4/VASA (AF2030, R and D Systems, Minneapolis, MN) and 1:250 for all other antibodies (STRA8, ab49405, Abcam, Cambridge, UK; STRA8, ab49405, Abcam; DMC1, sc227268, Santa Cruz Biotechnology, Dallas, TX; SCP3, sc-74569, Santa Cruz; γ-H2AX (Ser139) clone JBW301, 05–636, Millipore Sigma, Burlington, MA). Slides were then incubated with secondary antibodies conjugated to fluorophores (donkey anti-goat Alexa Fluor 488, donkey anti-rabbit RRX, and donkey anti-mouse Cy5, all from Jackson ImmunoResearch) at a 1:250 dilution. Slides were counterstained with DAPI prior to mounting with VECTASHIELD Antifade Mounting Medium for Fluorescence (Vector Laboratories, Burlingame, CA).

For colorimetric detection of STRA8, slides were treated with 3% hydrogen peroxide for 10 min, blocked with 2.5% horse serum (Vector Laboratories), and incubated for 30 min at room temperature with primary antibodies (ab49405, Abcam; ab49602, Abcam) diluted 1:500 in 2.5% horse serum. The slides were then washed with PBS, incubated with ImmPRESS peroxidase-conjugated secondary antibodies (Vector Laboratories), and incubated with DAB substrate (Vector Laboratories). For detection with a mouse antibody, the Mouse on Mouse ImmPRESS kit (Vector Laboratories) was used following manufacturer's instructions, with a 1:200 dilution of the FLAG M2 antibody (F1804, MilliporeSigma). Slides were counterstained with periodic acid-Schiff (MilliporeSigma), and/or with Mayer's hematoxylin (Thermo Fisher Scientific). Slides were dehydrated and mounted with Permount (Thermo Fisher Scientific).

## Spermatogenesis synchronization

Synchronization of spermatogenesis for 2S and 3S was performed following a synchronization protocol (*Hogarth et al., 2013*) that was subsequently modified (*Romer et al., 2018*). Briefly, male mice were injected daily from postnatal day (P) two to P8 with WIN18,446 (Santa Cruz Biotechnology; 0.1 mg/gram body weight) and on P9 with retinoic acid (RA) (MilliporeSigma; 0.0125 µg/gram body weight). All injections were performed subcutaneously over the shoulders. Mice were euthanized ~6.75 days (d) after the RA injection to obtain testes enriched for preleptotene cells. Each pair of synchronized testes yielded roughly five million cells. From each mouse, a small tissue biopsy was reserved for histology to confirm proper enrichment of preleptotene stage spermatocytes, and the rest was used for ChIP-seq or cell sorting followed by RNA-seq.

To quantify the extent of preleptotene cell enrichment, testis tissue was fixed in Bouin's solution for 3 hr. Testis sections were stained with STRA8 antibody (Abcam ab49405) and counter-stained with hematoxylin. For each sample, all cells in five microscope fields (100X magnification) were counted and classified according to morphology (*Russell, 1990*) and STRA8 expression status. The fraction of all cells that are STRA8-expressing preleptotene cells was obtained by dividing the number of preleptotene cells with robust STRA8 staining by the total number of cells (somatic and germ cells combined).

## Estimating the fraction of cells initiating meiosis in the adult testis

The fraction of all cells in the unperturbed adult testis that are STRA8-expressing preleptotene spermatocytes was estimated using known parameters for testis cell-type composition and the duration of various states within germ-cell development. First, the total number of cells in an adult mouse testis was estimated by summing the numbers of individual testis cell types (from *Tegelenbosch and de Rooij, 1993*: $3 \times 10^6$ Sertoli cells, $2.8 \times 10^6$ spermatogonia, $25 \times 10^6$ spermatocytes, $99 \times 10^6$ spermatids; from *Vergouwen et al., 1993*: $3.5 \times 10^6$ interstitial cells), yielding $133.3 \times 10^6$ cells per testis. Spermatocytes comprise 18.8% of this total.

Next, the proportion of preleptotene spermatocytes among all spermatocytes was estimated by noting that, in an adult testis, this proportion is approximated by the length of time a cell spends as a preleptotene spermatocyte relative to the total time spent as a spermatocyte. The total

spermatocyte lifespan is 326 hr (*Oakberg, 1956*). Preleptotene cells are present in the later half of stage VI, all of stage VII, and the first half of stage VIII in the cycle of the mouse seminiferous epithelium (*Russell, 1990*). Given the duration of these three stages (18.1 hr, 20.6 hr, and 20.8 hr, respectively (*Oakberg, 1956*)), the lifespan of a preleptotene spermatocyte is estimated to be ~40 hr. Thus, preleptotene cells comprise 40/326 = 12.3% of all spermatocytes and 2.3% of all cells in the testis, resulting in $\sim 3.1 \times 10^6$ preleptotene cells per testis.

Finally, the proportion of cells expressing STRA8 among all preleptotene cells was estimated. STRA8-positive spermatocytes are found in 0% of stage VI tubules, 71% of stage VII tubules, and 100% of stage VIII tubules (*Endo et al., 2015*). The duration of STRA8 expression can be estimated as (0.71 × (length of stage VII)) + (0.5 × (length of stage VIII)) = ~25 hr. Thus, STRA8-positive cells represent 25/40 = 62.5% of all preleptotene cells. The total number of STRA8-expressing preleptotene cells is therefore $0.625 \times 3.0 \times 10^6 = 1.88 \times 10^6$ per testis. We use these numbers to estimate that STRA8-expressing preleptotene spermatocytes comprise $1.88 \times 10^6 / 133.3 \times 10^6 = \sim 1.4\%$ of all cells in the adult testis.

## Isolation of sorted preleptotene cells

Purified preleptotene cell populations for RNA-seq were obtained following the '3S' protocol (*Romer et al., 2018*). With this system, the estimated purity of sorted preleptotene cells is ~90%. Germ-cell lineage sorting was achieved with the $Ddx4^{Cre}$ and $Rosa26^{tdTomato}$ alleles: $Rosa26^{tdTomato}$ contains a loxP-STOP-loxP-tdTomato construct at the endogenous $Rosa26$ locus, such that the STOP codon is excised in the presence of Cre recombinase. When combined with the $Ddx4^{Cre}$ allele, the $Rosa26^{tdTomato}$ allele yields mice with tdTomato expression only in germ cells. We mated $Ddx4^{Cre/+}Stra8^{+/-}$ mice with $Rosa26^{tdTomato/tdTomato}$ $Stra8^{+/-}$ mice, and we then treated their male offspring with WIN18,446/RA to synchronize spermatogenesis. We used synchronized testes from $Ddx4^{Cre/+}Rosa26^{tdTomato/+}$ mice that were either heterozygous ($Stra8^{+/-}$) or homozygous ($Stra8^{-/-}$) for the $Stra8$-deficient allele. Testes were collected ~6.75 d after injecting RA and reduced to a single-cell suspension using collagenase, trypsin, and DNaseI. DAPI was added to cells prior to cell sorting on the FACSAria SORP (BD Biosciences, San Jose, CA). Single cells were first gated based on light scatter (forward and side scatter), and high DAPI fluorescence (dead) cells were excluded. We then used tdTomato fluorescence intensity to sort only the preleptotene stage spermatocytes.

## Cell culture

v6.5 embryonic stem cells (ESCs) were cultured at 37°C in 5% $CO_2$ in Dulbecco's modified Eagle's medium supplemented with 15% fetal bovine serum, 2 mM L-glutamine, 0.1 mM non-essential amino acids, 1% penicillin/streptomycin, 0.1 mM β–mercaptoethanol, and 1,000 U/mL leukemia inhibitory factor (ESGRO, MilliporeSigma). Cells were initially cultured on mitocyin C-treated CF6Neo mouse embryonic fibroblasts (MTI-GlobalStem). To induce STRA8 expression, ESCs were transferred to gelatin-coated culture dishes and treated with culture medium containing 10 µM retinoic acid dissolved in DMSO. For an uninduced negative control, an equivalent volume of DMSO was added to the culture medium.

## Immunoblotting

To prepare protein lysates, testes (snap frozen in liquid nitrogen and subsequently thawed) or ESCs were homogenized in lysis buffer [50 mM HEPES (pH 7.4), 1 mM EGTA, 1 mM $MgCl_2$, 100 mM KCl, 10% glycerol, 0.05% NP-40] supplemented with EDTA-free protease inhibitor (Roche Diagnostics, Risch-Rotkreuz, Switzerland) and Benzonase nuclease (MilliporeSigma), incubated at 4°C with rotation for 30 min, and then centrifuged at 20000 g for 15 min at 4°C. The soluble fraction was used as the lysate. Proteins were denatured in sample buffer for 10 min at 70°C, resolved on a NuPAGE 4–12% Bis-Tris gel (Thermo Fisher Scientific), and transferred to a nitrocellulose membrane. The membrane was blocked in 5% BSA/Tris-buffered saline containing 0.1% Tween-20 (TBST) for 1 hr at room temperature and then incubated with a primary antibody solution prepared in 5% BSA/TBST. STRA8 antibodies (ab49602 or ab49405, Abcam) were used at a 1:1000 dilution with an overnight incubation at 4°C, followed by a 1 hr room-temperature incubation with a 1:5000 dilution of a peroxidase-conjugated anti-rabbit secondary antibody (Jackson ImmunoResearch). Primary antibody incubation with FLAG-HRP (1:1000; A8592, MilliporeSigma) was performed overnight at 4°C. Incubation with

alpha tubulin-HRP (1:5000; ab40742, Abcam) was performed at room temperature for 1 hr. Proteins on the membranes were detected by addition of Lumi-Light Western Blotting Substrate (Roche).

## Chromatin immunoprecipitation

ChIP-seq was performed using a modification of a small-scale ChIP protocol described previously (*Lesch et al., 2013*). Synchronized testes were dissected into PBS and dissociated into a single-cell suspension in 0.25% trypsin + 0.1 mM EDTA with collagenase (type 1) (Worthington Biochemical, Lakewood, NJ) and 0.1% hyaluronidase (MilliporeSigma). Dissociated cells were washed in PBS and crosslinked for 8 min in 1% formaldehyde at room temperature. Crosslinking was quenched by addition of glycine (to a concentration of 0.125 M) for 5 min at room temperature. Cell pellets were frozen at −80°C. To perform ChIP, frozen cell pellets were thawed on ice for 5 min and resuspended in 100 μL of ChIP lysis buffer [1% SDS, 10 mM EDTA, 50 mM Tris-HCl (pH 8.1)]. ChIP dilution buffer [0.01% SDS, 1.1% (vol/vol) Triton X-100, 1.2 mM EDTA, 16.7 mM Tris-HCl (pH 8.1), 167 mM NaCl] was then added to achieve a total volume of 300 μL. Samples were sonicated in a Bioruptor (Diagenode, Liege, Belgium) for 12 min, 30 s on/30 s off. For each, the volume was then brought up to 1 mL with dilution buffer plus protease inhibitors, and 50 μL of this reaction was set aside as an input chromatin control. To prepare antibody-bound beads for ChIP, 25 μL of Dynabeads Protein G (Thermo Fisher) were washed three times in block solution (0.5% BSA in PBS), resuspended in 100 μL of block solution, and incubated with antibody [2.5 μg of FLAG M2 antibody (F1804, MilliporeSigma)] for 8 hr at 4°C with rotation. These beads were then washed three times in 200 μL of block solution, resuspended in 50 μL dilution buffer, and added to sonicated chromatin. The immunoprecipitation reactions were incubated overnight, rotating, at 4°C. After overnight incubation, beads were washed two times in 700 μL low-salt immune complex wash buffer [0.1% SDS, 1% Triton X-100, 2 mM EDTA, 20 mM Tris-HCl (pH 8.1), 150 mM NaCl], two times in LiCl wash buffer [0.25 M LiCl, 1% Nonidet P-40, 1% deoxycholate, 1 mM EDTA, 10 mM Tris·HCl (pH 8.1)], and two times in TE (10 mM Tris-HCl, 1 mM EDTA, pH 8.0). Unbound fractions were used to evaluate fragmentation. Bound chromatin was eluted twice (for 10 min each elution) in 125 μL elution buffer (0.2% SDS, 0.1 M NaHCO$_3$, 5 mM DTT in TE) at 65°C. Crosslinking was reversed by incubation for 8 hr at 65°C with 400 rpm shaking. ChIP and input samples were treated with 0.2 mg/mL RNase A for 2 hr at 37°C and with 0.1 mg/mL proteinase K for 2 hr at 55°C. The DNA samples (from ChIP and input chromatin) were purified using the ChIP Clean and Concentrator kit (Zymo Research, Irvine, CA) and eluted in 13 μL of the kit's elution buffer.

ChIP-seq with the FLAG antibody was performed on three biological replicates for each condition (*Stra8$^{+/+}$* and *Stra8$^{FLAG/FLAG}$*) with ~5 million cells for each replicate. For ChIP-seq, we chose our sample size of three replicates before beginning our study, because this number exceeds the suggested ENCODE guideline of two replicates. To minimize batch-specific variation, each *Stra8$^{FLAG/FLAG}$* ChIP replicate was processed in parallel with a *Stra8$^{+/+}$* ChIP replicate: the two ChIP experiments were performed in parallel, the sequencing libraries (from the two samples and their corresponding two input controls) were prepared in parallel, and a pool of the four barcoded libraries was run on a single flow cell. The three biological replicates were processed in three different batches. Libraries for ChIP samples and input controls were prepared using the TruSeq ChIP Sample Preparation Kit (Illumina, San Diego, CA) (replicates 1 and 2) or the Accel-NGS 2S Plus DNA Library Kit (Swift Biosciences, Ann Arbor, MI) (replicate 3) according to the manufacturer's instructions. The sequencing libraries were sequenced with 40 bp or 80 bp single-end reads on an Illumina HiSeq 2500 machine.

## Data analysis: ChIP-seq

For ChIP-seq analysis, low-quality sequencing reads were filtered out using the FASTX-Toolkit v0.0.14 (http://hannonlab.cshl.edu/fastx_toolkit/index.html), and reads were trimmed to 40 bp. Remaining reads were aligned to the mm10 mouse genome using bowtie1 v1.2.0, allowing one mismatch per read (*Langmead et al., 2009*). For every ChIP-seq replicate, peaks of read density were identified using MACS2 v2.1.1.20160309 (*Zhang et al., 2008*) using the corresponding input chromatin as the control sample and the default *P*-value cutoff of $1\times10^{-5}$. Peaks overlapping blacklist regions (*Supplementary file 5*) were then removed. To determine whether peaks were associated with transcript start sites (TSSs), TSS coordinates were downloaded from Ensembl (release 90, GRCm38.p5), considering only protein-coding transcripts on the reference chromosomes belonging

to the 'GENCODE basic' annotation. The TSS for the germ-cell gene *Gcna* (*Carmell et al., 2016*) (Genbank accession number KX981576.1, TSS at chrX:101,695,571) was added. A peak was associated with a TSS if the summit of the peak (defined by MACS2) was located within one kilobase (kb) of the TSS (either upstream or downstream). If a peak summit was within 1 kb of multiple TSSs, the peak was assigned to all associated transcripts.

To identify a list of STRA8-bound genes, we first determined the genes whose promoters were associated with a STRA8 peak in each of the ChIP-seq replicates (*Stra8*$^{+/+}$ and *Stra8*$^{FLAG/FLAG}$). We filtered out genes whose promoters might be non-specifically bound by the antibody by removing genes identified in any of the three *Stra8*$^{+/+}$ ChIP seq replicates. Genes that were STRA8-bound in multiple *Stra8*$^{FLAG/FLAG}$ ChIP-seq replicates were identified by overlapping the lists of genes identified in individual replicates.

To quantify the ChIP-seq read density at gene promoters, potential PCR duplicates were first removed using the rmdup function in Samtools v1.3.1 (*Li et al., 2009*). The number of reads at the promoter (within 1 kb of the TSS) of each transcript was determined using htseq-count (*Anders et al., 2015*); transcript count values were summed to obtain a count for each gene. The counts were normalized to reads per million (rpm) by scaling to the total number of unduplicated reads in each dataset, and the normalized input chromatin read count was subtracted from the normalized ChIP-seq read count.

To visualize the average ChIP-seq density at the TSS of genes, we employed ngs.plot v2.61 (*Shen et al., 2014*). We removed all potential PCR duplicates from bam files prior to plotting. To show average ChIP-seq read density across all replicates, the bam files from all three replicates were concatenated into a single file before plotting. We supplied either a list of all genes, or the lists of genes identified as STRA8-bound in specified numbers of replicates.

To visualize the input-subtracted normalized ChIP-seq signal, we generated a bigWig file with the input chromatin reads subtracted from the ChIP sample reads using the bamcompare tool v2.5.3 from deepTools (*Ramírez et al., 2016*) (options: –ratio subtract, –normalizeUsingRPKM, –ignoreDuplicates, –smoothLength 100, and –binSize 10). The ChIP-seq signal was visualized using the mm10 genome assembly on the UCSC Genome Browser (http://genome.ucsc.edu/). The ChIP-seq tracks shown represent individual replicates where indicated, or pooled signal from all three *Stra8*$^{FLAG/FLAG}$ ChIP-seq samples.

To determine whether the promoters of STRA8-bound genes were enriched for CpG islands (CGIs), the coordinates of CpG islands in the mouse genome (mm10) were downloaded using the UCSC Table Browser (*Karolchik et al., 2004*) at http://genome.ucsc.edu/. A gene was considered to have a CGI promoter if a CGI overlapped the 2 kb window surrounding the TSS for at least one of its transcripts.

## RNA-seq sample preparation

For RNA extraction, preleptotene cells were sorted into 500 μL of PBS with 1% BSA, plus 3.5 μL of RNAseOut (Thermo Fisher). RNA extraction was performed using a QIAshredder homogenizer (Qiagen, Hilden, Germany) and RNeasy Plus Micro kit (Qiagen), and RNA was eluted in 14 μL of RNase-free water. Prior to library preparation, RNA was stored at −80°C so that all RNA-seq libraries could be prepared together. RNA-seq libraries were prepared with the TruSeq Stranded mRNA kit (Illumina) with poly-A selection. To avoid batch effects, all seven barcoded libraries were pooled together, and this pool was sequenced with 40 bp single-end reads across two flow-cell lanes on an Illumina HiSeq 2500 machine. For each library, sequencing reads from the two lanes were combined prior to data analysis.

## Data analysis: RNA-seq

For RNA-seq analysis, high-quality reads were first filtered using the FASTX-Toolkit (http://hannon-lab.cshl.edu/fastx_toolkit/index.html). Expression levels of all transcripts in the Ensembl release 90 annotation (GENCODE basic subset, with the addition of mouse *Gcna* [Genbank accession KX981576.1]) were estimated using Kallisto v0.43.0 (*Bray et al., 2016*) with sequence-bias correction. Expression levels for protein-coding transcripts on reference chromosomes (mm10) were extracted, summed by gene using tximport v1.6.0 (*Soneson et al., 2016*), and re-normalized to transcript-per-million (TPM) units. To calculate fold-changes in expression level between different

genotypes/conditions, the estimated count values from kallisto were supplied to DESeq2 v1.18.1 (*Love et al., 2014*) with default parameters. An FDR cutoff of $q < 0.05$ was used to determine statistically significant gene expression changes.

To calculate the degree of testis-biased expression for each gene, RNA-seq datasets from *Merkin et al. (2012)* were downloaded for nine adult mouse tissues: brain, colon, heart, kidney, liver, lung, skeletal muscle, spleen, and testis. Expression levels for all cDNA transcripts annotated in Ensembl release 84 were estimated with kallisto. Genes from Ensembl release 84 were matched to genes in release 90 by Ensembl gene ID. The expression level of each gene in each tissue was estimated as its median expression level (in TPM units) across three individuals. A gene was considered to be testis-biased if its expression level in the testis was greater than its summed expression level across the eight remaining tissues (i.e., TPM in the testis > 0.5 of summed TPM values across all tissues).

## Data analysis: ENCODE ChIP-seq datasets

To identify potential enhancer sites, we downloaded replicated H3K4me1 peaks from adult mouse testis from the ENCODE Consortium (*ENCODE Project Consortium, 2012*) and removed peaks that overlap blacklist regions. H3K4me1 peaks were called as enhancer sites if they did not overlap the promoters of protein-coding genes, which were defined as the 2 kb window surrounding the TSS. TSS coordinates were the same as those in the STRA8 ChIP-seq analysis.

To identify cell-cycle transcription factor targets, ENCODE ChIP-seq data for E2F4 (mouse), E2F1 (human), and FOXM1 (human) were used. To obtain a high-confidence list of target genes, we downloaded the coordinates of IDR-thresholded peaks (optimal or pseudoreplicated), removed any peaks that overlapped blacklist regions, and identified the transcripts whose promoters overlapped the summits of the ChIP-seq peaks. For human, all hg19 genome coordinates were first converted to GRCh38 coordinates using the UCSC Genome Browser LiftOver tool. Peaks were mapped to transcripts/genes using TSS coordinates downloaded from Ensembl release 90 (GRCh38.p10), limiting the analysis to protein-coding transcripts on the reference chromosomes in the 'GENCODE basic' annotation. One-to-one mouse orthologs of human genes were identified using the BioMart database (*Smedley et al., 2015*) (Ensembl release 90). See *Supplementary file 5* for the list of ENCODE datasets used.

## Gene list analysis

Gene Ontology analysis was performed with PANTHER over-representation test v13.1 (*Mi et al., 2017*) to identify biological processes that are enriched among the 1,351 STRA8-activated genes compared to a background list of 12,545 preleptotene-expressed genes (all genes expressed at TPM > 1 in at least one of the seven preleptotene samples). *P*-values for overrepresentation were calculated using one-tailed binomial tests with Bonferroni correction. To determine whether non-meiotic cell cycle genes were overrepresented, a custom Gene Ontology category was generated by subtracting all terms that mapped to the process 'meiotic cell cycle' (GO: 0051321) from the terms that mapped to the process 'cell cycle' (GO: 0007049). To determine whether STRA8-activated genes are enriched for representative cell-cycle genes, we used the one-to-one mouse orthologs of the human cell-cycle genes in Table S2 from *McKinley and Cheeseman (2017)*. The *P*-value for this enrichment was calculated using a one-tailed hypergeometric test.

## Motif analysis

DNA motifs enriched in target genomic regions were identified using the HOMER motif discovery software package v4.9.1 (*Heinz et al., 2010*) with the mm10 genome configuration v5.10. Motif scores were calculated using the binomial distribution. To identify motifs enriched at the STRA8 binding sites, we first pooled all sequencing reads from the three *Stra8^{FLAG/FLAG}* ChIP-seq samples, identified peaks of binding using MACS2 (*Zhang et al., 2008*) v2.1.1.20160309 at a *P*-value cutoff of $< 1 \times 10^{-5}$, and then identified all peak summits that were at the promoter (within 1 kb of the TSS) of the 1,351 STRA8-activated genes. The findMotifsGenome tool was used to identify motifs de novo that were enriched among the 100 bp windows surrounding these summits (50 bp on each side of the summit); for this analysis, the background set consisted of 100 bp sequences that were matched for GC-content. For identification of motifs enriched near the TSS (−200 bp to +200 bp) of

the STRA8-activated genes, the findMotifs.pl tool was used, using a target gene list of the 1,351 STRA8-activated genes and a background list of all other genes expressed in preleptotene samples. To determine the number of motifs at a gene's promoter, we used the scanMotifGenomeWide.pl tool to determine all locations of the motif across the genome and counted the number of motif instances at the promoter (2 kb surrounding the TSS) for each annotated transcript. For genes associated with multiple transcripts, we used the motif count from the transcript with the most promoter motifs. To determine the number of promoter motifs at human genes, one-to-one human orthologs of mouse genes were identified using the BioMart database (*Smedley et al., 2015*) (Ensembl release 90).

## Statistical tests

Statistical tests were conducted using RStudio v1.1.414 as described here, unless otherwise noted. Overlaps of gene lists were computed using one-tailed hypergeometric tests. Analysis of contingency tables was conducted using Fisher's exact tests. One- or two-sided Mann-Whitney $U$ tests (Wilcoxon rank sum tests) were used to determine if values from one sample are higher or lower than values from a second sample. Correlations among ChIP-seq and RNA-seq biological replicates were assessed with the Pearson correlation coefficient. The correlation between motif count and ChIP-seq score was calculated using the Spearman correlation coefficient; the $P$-value for this correlation was obtained by permutation testing. All box plots show sample medians and interquartile ranges (IQRs), with whiskers extending no more than $1.5 \times$ IQR and outliers suppressed.

## Phylogenetic analysis

Nucleotide and amino acid percent identities were calculated using BLAST (NCBI). Phylogenetic trees were constructed using open reading frame sequences. Sequences used for the tree are available in *Supplementary file 4*. Maximum likelihood tree lengths were calculated using the DNAML program from the PHYLIP package, version 3.66.

## Data availability

The ChIP-seq and RNA-seq data generated in this study are available at NCBI Gene Expression Omnibus (accession number GSE115928). Gene lists generated in this study, including lists of genes differentially expressed at meiotic initiation, STRA8-bound genes, and STRA8-activated genes are available in *Supplementary file 1*. RNA-seq results and STRA8 binding status for all protein-coding genes are available in *Supplementary file 2*, as are the numbers of CNCCTCAG promoter motifs for all genes. Data for a meiotic prophase gene list described previously (*Soh et al., 2015*) are available in *Supplementary file 3*.

## Acknowledgments

We thank H Yang, H Schorle, and M Goodheart for help with generating the *Stra8*[FLAG] and *Stra8*-[Δ121]alleles; K Romer, H Christensen, and M Mikedis for help with mouse injections; H Skaletsky for generation of the phylogenetic tree; and S Naqvi for help with analysis of published RNA-seq data. We are grateful to the Page Lab for valuable feedback, and W Bellott, H Christensen, C Edwards, A Godfrey, J Hughes, M Mikedis, P Nicholls, and T Orr-Weaver for comments on the manuscript. We thank Abcam for disclosing immunogenic peptide sequences, the Whitehead Genome Technology Core for library preparation and Illumina sequencing, and the Whitehead Institute FACS Facility for cell sorting. Microscopy was conducted in part at the WM Keck Foundation Biological Imaging Facility at the Whitehead Institute. This work was supported by a National Science Foundation Graduate Research Fellowship (to MLK) and the Howard Hughes Medical Institute.

## Additional information

### Funding

| Funder | Grant reference number | Author |
| --- | --- | --- |
| Howard Hughes Medical Institute | Investigator | David C Page |

| National Science Foundation | Graduate Research Fellowship | Mina L Kojima |

The funders had no role in study design, data collection and interpretation, or the decision to submit the work for publication.

## Author contributions

Mina L Kojima, Conceptualization, Data curation, Formal analysis, Validation, Investigation, Visualization, Methodology, Writing—original draft, Writing—review and editing; Dirk G de Rooij, Formal analysis, Investigation, Writing—review and editing; David C Page, Conceptualization, Resources, Supervision, Funding acquisition, Writing—original draft, Writing—review and editing

## Author ORCIDs

Mina L Kojima (iD) http://orcid.org/0000-0002-3387-8608
Dirk G de Rooij (iD) https://orcid.org/0000-0003-3932-4419
David C Page (iD) http://orcid.org/0000-0001-9920-3411

## Ethics

Animal experimentation: This study was performed in strict accordance with the recommendations in the Guide for the Care and Use of Laboratory Animals of the National Institute of Health. All animals were handled and experiments conducted according to approved institutional animal care and use committee (IACUC) protocols (#0617-059-20) of the Massachusetts Institute of Technology.

## Decision letter and Author response

Decision letter https://doi.org/10.7554/eLife.43738.056
Author response https://doi.org/10.7554/eLife.43738.057

# Additional files

## Supplementary files

• Supplementary file 1. Relevant gene lists generated by this study.
DOI: https://doi.org/10.7554/eLife.43738.027

• Supplementary file 2. STRA8 ChIP-seq status, RNA-seq data, and CNCCTCAG motif count for all protein-coding genes.
DOI: https://doi.org/10.7554/eLife.43738.028

• Supplementary file 3. STRA8 ChIP-seq status and RNA-seq data for all meiotic prophase genes listed in *Soh et al. (2015)*.
DOI: https://doi.org/10.7554/eLife.43738.029

• Supplementary file 4. Sequences used to generate the *Taf7l2* phylogenetic tree.
DOI: https://doi.org/10.7554/eLife.43738.030

• Supplementary file 5. ENCODE datasets used in this study.
DOI: https://doi.org/10.7554/eLife.43738.031

• Supplementary file 6. Primer and oligonucleotide sequences used in this study.
DOI: https://doi.org/10.7554/eLife.43738.032

• Transparent reporting form
DOI: https://doi.org/10.7554/eLife.43738.033

## Data availability

The ChIP-seq and RNA-seq data generated in this study are available at NCBI Gene Expression Omnibus (accession number GSE115928). Gene lists generated in this study, including lists of genes differentially expressed at meiotic initiation, STRA8-bound genes, and STRA8-activated genes are available in Supplementary file 1. RNA-seq results and STRA8 binding status for all protein-coding genes are available in Supplementary file 2, as are the numbers of CNCCTCAG promoter motifs for

all genes. Data for a meiotic prophase gene list described previously (Soh et al., 2015) are available in Supplementary file 3. Source data files have been provided for Figures 1-6.

The following dataset was generated:

| Author(s) | Year | Dataset title | Dataset URL | Database and Identifier |
|---|---|---|---|---|
| Kojima ML, Page DC | 2018 | Characterization of molecular changes at meiotic initiation in mice | https://www.ncbi.nlm.nih.gov/geo/query/acc.cgi?acc=GSE115928 | NCBI Gene Expression Omnibus, GSE115928 |

The following previously published datasets were used:

| Author(s) | Year | Dataset title | Dataset URL | Database and Identifier |
|---|---|---|---|---|
| Merkin JJ, Burge CB | 2012 | Evolutionary dynamics of gene and isoform regulation in mammalian tissues | https://www.ncbi.nlm.nih.gov/geo/query/acc.cgi?view=brief&acc=GSE41637 | NCBI Gene Expression Omnibus, GSE41637 |
| Ren B | 2012 | H3K4me1 ChIP-seq on 8-week mouse testis | https://www.encodeproject.org/files/ENCFF449YGK/ | ENCODE, ENCSR000CCV |
| Snyder M | 2011 | E2F4 ChIP-seq on mouse CH12 produced by the Snyder lab | https://www.encodeproject.org/files/ENCFF486IVR/ | ENCODE, ENCSR000ERU |
| Snyder M | 2011 | E2F4 ChIP-seq on mouse MEL produced by the Snyder lab | https://www.encodeproject.org/files/ENCFF624HQU/ | ENCODE, ENCSR000ETY |
| Wold B | 2011 | E2F4 ChIP-seq on mouse C2C12 differentiated for 60 hours | https://www.encodeproject.org/files/ENCFF067KOQ/ | ENCODE, ENCSR000AII |
| Snyder M | 2016 | E2F1 ChIP-seq on human K562 | https://www.encodeproject.org/files/ENCFF445VTT/ | ENCODE, ENCSR563LLO |
| Farnham P | 2011 | E2F1 ChIP-seq on human HeLa-S3 | https://www.encodeproject.org/files/ENCFF002CSG/ | ENCODE, ENCSR000EVJ |
| Snyder M | 2017 | FOXM1 ChIP-seq on human K562 | https://www.encodeproject.org/files/ENCFF778PWE/ | ENCODE, ENCSR429QPP |
| Snyder M | 2017 | FOXM1 ChIP-seq on human HEK293T | https://www.encodeproject.org/files/ENCFF685TME/ | ENCODE, ENCSR831EIW |

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

## Appendix 1

DOI: https://doi.org/10.7554/eLife.43738.034

## Domains of the STRA8 protein

STRA8 was previously hypothesized to be a basic helix-loop-helix (bHLH) transcription factor, based on the presence of a stretch of basic amino acids followed by an HLH domain (*Baltus et al., 2006*; *Tedesco et al., 2009*) in amino acids (aa) 28-79 of the 393-aa STRA8 isoform. bHLH domains are found in transcription factors. Their HLH regions enable dimerization (either hetero-dimerization or homo-dimerization) with other HLH proteins, and this is required for protein activity. The basic residues preceding the HLH mediate DNA binding. Beyond the presence of this putative domain, there had been no *in vivo* evidence that STRA8 functions as a bHLH transcription factor. However, the region containing the bHLH domain is likely to be important for STRA8 function, as it is among the most highly conserved regions of the protein (*Baltus et al., 2006*). In addition, *in vitro* studies using cell lines transfected with STRA8 revealed that the first 84 amino acids (which includes the bHLH domain) were required for nuclear localization; nuclear localization was also impaired when arginines in the basic region were mutated to alanine residues (*Tedesco et al., 2009*). We now find *in vivo* that a STRA8 isoform that lacks the first 111 amino acids does not localize to the nucleus of preleptotene cells; germ cells that express only this shorter isoform of STRA8 are not able to initiate meiosis (*Figure 2—figure supplement 3*). These findings point to a critical role for the N terminus — presumably the HLH domain — in STRA8's nuclear localization *in vivo* and its function in meiotic initiation.

The mouse STRA8 amino acid sequence has a long stretch of glutamic acid residues (amino acids 143-193 of the longer isoform). However, this stretch of glutamic acids is not well conserved among different species.

STRA8 may have another DNA binding domain in addition to the bHLH domain. Using the Phyre2 web portal (*Kelley et al., 2015*), which predicts domains and secondary structures from amino acid sequences, we found that that amino acids ~216-268 of the 393-aa STRA8 protein likely encode a high-mobility group (HMG) box domain. This domain is often used by transcription factors for DNA binding but has not yet been studied in the context of the STRA8 protein.

