## [Decision Letter]

Thank you for submitting your article "Amplification of a broad transcriptional program by a common factor triggers the meiotic cell cycle in mice" for consideration by *eLife*. Your article has been reviewed by three peer reviewers, including Bernard de Massy as the Reviewing Editor and Reviewer #1, and the evaluation has been overseen by Marianne Bronner as the Senior Editor. The following individual involved in review of your submission has also agreed to reveal their identity: Kei-ichiro Ishiguro (Reviewer #3).

The reviewers have discussed the reviews with one another and the Reviewing Editor has drafted this decision to help you prepare a revised submission.

Summary:

In this study, Kojima, de Rooij and Page present an impressive piece of work related to the role of *Stra8*. Since its discovery twelve years ago STRA8 is the enigmatic gatekeeper of the mitotic/meiotic switch in vertebrates, though its molecular function remained unclear. In this study, the authors addressed the molecular function of STRA8 in the initiation of meiosis. The authors' comprehensive analyses with RNA-seq and ChIP-seq using spermatogenesis synchronization method lead the authors to conclude that STRA8 binds and activates promoters of germ-line specific genes for initiation of meiosis and those of G1-S cell cycle regulation. Overall manuscript is well written and provides insight into meiotic initiation in mouse.

Essential revisions:

However the presentation of the data and its interpretation requires clarification. The main issues that require clarification are:

1) The authors should define more accurately what they define as meiotic initiation and also as preleptotene stage. It seems that one of their conclusions is that meiotic initiation is before/at/during (some ambiguity here) preleptotene. In addition, they have to take into account and integrate into their study the observation that Mark et al., 2008 reported meiotic initiation in *Stra8* deficient males. How come this study is not even referenced? For instance one can read the ambiguity in the sentence: "These observations suggest that the decision to initiate meiosis occurs in the preleptotene stage, upstream of meiotic S phase": How can initiation occur in preleptotene and upstream of S phase, if preleptotene is S phase?

2) The CHipSeq analysis is performed on 2S cells which includes B type spermatogonia and only 44% Stra positive preleptotene cells. Romer et al. report in some samples 20% B type spermatogonia and also *Stra8* negative preleptotene cells (based on immuno-detection). What was the composition of the 2S samples used for ChIPSeq? Some *Stra8* bound sites could be genes regulated in these cells but potentially in spermatogonia. This should be clarified, discussed and it could indicate a role for *Stra8* before meiotic initiation. Ideally, the detection of *Stra8* binding in purified spermatogonia could solve this issue.

3) It is important to validate the use of the *Stra8^FLAG^* allele: Are *Stra8^FLAG^* homozygous mice fully fertile and is the testis transcriptome same as wild type?

4) The authors should analyze and present the comparison between *Stra8* bound genes, genes upregulated in the presence of *Stra8* (compared to *Stra8* KO) and genes upregulated at or before meiotic initiation in wild type, in a more synthetic way. This does not exclude to have specific sections for meiotic prophase and cell cycle genes. In addition, the authors should stress that these RNAseq analysis have a limitation: they do not detect other regulatory levels such as RNA stability and translation. Recent papers have shown this to be very important during meiosis.

In the paper, the tables provided with the list of genes deregulated in *Stra8^-/-^* male preleptotene, ovaries, testis-specific and bound by STRA8FLAG is actually difficult to follow. A unified table with gene ID and when possible the alias (e.g. KASH5, IHO1, SIX6OS1) would be very helpful to readers especially to identify meiotic actors.

Other comments:

5) It would be interesting to know if *Stra8* has on its own transcription activation activity. This could be tested by a transfection assay.

6) *Stra8* binding to its own promoter: The authors previously found that down-regulation of *Stra8* gene expression depends on STRA8 itself. There is some apparent contradiction here. Could *Stra8* have a negative effect on its expression level?

7) Explain the two categories of high and low *Stra8*: What is responsible for this variation? Please define the "low STRA8" are these spermatogonia, early pre-leptotene, or leptotene cells?

8) The normalization procedure for RNA seq requires further investigation: it seems that RPMs have been used to normalize. Is this correct? What are the expression levels of housekeeping genes (wt vs. *Stra8* KO)? Could these be used to normalize?

Why is *Rec8* down-regulated? Could this result from the normalization used?

9) The authors should discuss what the down-regulated genes are and whether this is compatible with a direct and/or indirect effect.

10) Figure 1—figure supplement 2A: include the *Stra8* KO control.

11) This sentence is not very convincing: “Even genes that were bound in only one replicate are likely true STRA8 targets; their promoters display appreciable ChIP-seq signal compared to non-target genes (Figure 2—figure supplement 4G).” What is the meaning of "appreciable"? Either revise or delete.

12) What is the chromatin status of *Stra8* bound promoters? A survey and analysis of available data for H3K4me3, H3K9ac, H3K27me3… should be performed. This could provide insight as to whether *Stra8* is actually promoting initiation of transcription or release from pausing.

13) In the Introduction, the authors mention the issue of the genetic background on the *Stra8* mutant phenotype. Have they looked at whether some polymorphism between 129 and B6 could be detected at *Stra8* binding sites? More generally what is the evolutionary conservation (across mammals and vertebrates) of *Stra8* motifs at promoters?

14) Figure 1E needs a better presentation: the plot indicates 12545 genes, but the two boxes, pink and green, cover all of them and extend to a Log change of zero? Is it possible to visualize the 2278 and 2361 genes within this plot?

15) Which database was used for the motifs bound by other transcription factors? The overlap with TCPAP2E is intriguing. What is the occurrence of TCPAP2E motifs over the promoters of *Stra8* up and down-regulated genes?

---

## [Author Response]

Essential revisions:However the presentation of the data and its interpretation requires clarification. The main issues that require clarification are:1) The authors should define more accurately what they define as meiotic initiation and also as preleptotene stage. It seems that one of their conclusions is that meiotic initiation is before/at/during (some ambiguity here) preleptotene. In addition, they have to take into account and integrate into their study the observation that Mark et al., 2008 reported meiotic initiation in Stra8 deficient males. How come this study is not even referenced? For instance one can read the ambiguity in the sentence: "These observations suggest that the decision to initiate meiosis occurs in the preleptotene stage, upstream of meiotic S phase": How can initiation occur in preleptotene and upstream of S phase, if preleptotene is S phase?

We have revised the Introduction (third paragraph) to clarify the concepts of meiotic initiation and the preleptotene stage.

Preleptotene spermatocytes arise from the division of B spermatogonia and indeed undergo meiotic S phase. However, during the ~40 hour-long preleptotene stage, a long G1 phase precedes meiotic S phase: preleptotene cells arise in seminiferous cycle stage VI but do not initiate meiotic S phase until stage VII. Furthermore, robust STRA8 expression is not observed until stage VII, during the last ~25 hours of the preleptotene stage. These observations are consistent with meiotic initiation occurring *during* the preleptotene stage. (Please see Materials and methods for calculations.)

In the present manuscript we characterize STRA8’s function in preleptotene cells, because a strict *Stra8* requirement for male germ cells to initiate meiosis and progress beyond the preleptotene stage was demonstrated on the pure C57BL/6 genetic background (Anderson et al., 2008). We have edited the Introduction (third paragraph) to reflect Mark and colleagues’ findings on a mixed genetic background [C57BL/6 (50%) 129/Sv (50%)], in which *Stra8*-deficient male germ cells can initiate meiosis but fail to complete meiotic prophase I.

2) The CHipSeq analysis is performed on 2S cells which includes B type spermatogonia and only 44% Stra positive preleptotene cells. Romer et al. report in some samples 20% B type spermatogonia and also Stra8 negative preleptotene cells (based on immuno-detection). What was the composition of the 2S samples used for ChIPSeq? Some Stra8 bound sites could be genes regulated in these cells but potentially in spermatogonia. This should be clarified, discussed and it could indicate a role for Stra8 before meiotic initiation. Ideally, the detection of Stra8 binding in purified spermatogonia could solve this issue.

*Stra8* regulates not only meiotic initiation but also spermatogonial differentiation, and it is also expressed in A type spermatogonia (Endo et al., 2015). However, our STRA8 ChIP-seq experiments were designed to specifically shed light on STRA8’s role at meiotic initiation. We carefully chose the 2S testis samples for ChIP-seq, through STRA8 staining and histological staging, to ensure that most of STRA8 ChIP-seq signal derives from cells at meiotic initiation:

1) Of all cells that express STRA8 in our 2S samples used for ChIP-seq, ~88% are initiating meiosis and ~12% are spermatogonia.

2) Preleptotene cells express at least two-fold higher levels of STRA8 protein than spermatogonia (see STRA8 IHC in Figure 2A).

3) Therefore, at least ~88% of STRA8 ChIP-seq signal derives from cells initiating meiosis. Assuming that STRA8 levels in preleptotene cells are twice the levels in spermatogonia, ~94% of the STRA8 ChIP-seq signal comes from meiotic initiation.

We note that in Romer et al., 2018, we demonstrated the range of cell types enriched at 6.75-7 days following the Win18,446/RA synchronization procedure. Due to mouse-to-mouse variability in synchronization, each sample used must be carefully staged, as we have done here before selecting samples for ChIP-seq.

We provide the composition of the 2S samples used for ChIP-seq in Figure 2—figure supplement 4, and we have modified the main text (subsection “Identifying the targets of a meiotic initiation factor”, fourth paragraph) to alert readers to this data. We have also modified the text (see the third paragraph of the aforementioned subsection) to indicate that most of the STRA8 ChIP-seq signal comes from cells initiating meiosis.

3) It is important to validate the use of the Stra8^FLAG^ allele: Are Stra8^FLAG^ homozygous mice fully fertile and is the testis transcriptome same as wild type?

Both male and female *Stra8^FLAG/FLAG^*mice are fully fertile; we mated *Stra8^FLAG/FLAG^*homozygous mice to each other to produce *Stra8^FLAG/FLAG^*mice, as noted in Materials and methods, under the subsection “Mice”. The normal histology of adult *Stra8^FLAG/FLAG^*ovaries and testes is demonstrated in Figure 2—figure supplement 1D, F. We have revised the text (subsection “Identifying the targets of a meiotic initiation factor”, first paragraph) to make it clear that *Stra8FLAG/FLAG* mice have normal fertility and normal gonad histology. We have not investigated the testis transcriptome of *Stra8^FLAG/FLAG^*mice.

4) The authors should analyze and present the comparison between Stra8 bound genes, genes upregulated in the presence of Stra8 (compared to Stra8 KO) and genes upregulated at or before meiotic initiation in wild type, in a more synthetic way. This does not exclude to have specific sections for meiotic prophase and cell cycle genes. In addition, the authors should stress that these RNAseq analysis have a limitation: they do not detect other regulatory levels such as RNA stability and translation. Recent papers have shown this to be very important during meiosis.In the paper, the tables provided with the list of genes deregulated in Stra8^-/-^ male preleptotene, ovaries, testis-specific and bound by STRA8FLAG is actually difficult to follow. A unified table with gene ID and when possible the alias (e.g. KASH5, IHO1, SIX6OS1) would be very helpful to readers especially to identify meiotic actors.

We have modified Supplementary file 2 and 3 to make the data more accessible to readers. These modified files now include gene aliases.

We have edited the text (Discussion, second to last paragraph) to acknowledge that our data do not provide insight into post-transcriptional mechanisms.

Other comments:5) It would be interesting to know if Stra8 has on its own transcription activation activity. This could be tested by a transfection assay.

We have modified the text (subsection “Identifying the targets of a meiotic initiation factor”, first paragraph) to include a note about STRA8’s transcription activation activity: Tedesco et al., 2009 reported that, in cultured HEK293 cells, mouse STRA8 fused to a GAL4 DNA binding domain will activate transcription of a reporter gene.

6) Stra8 binding to its own promoter: The authors previously found that down-regulation of Stra8 gene expression depends on STRA8 itself. There is some apparent contradiction here. Could Stra8 have a negative effect on its expression level?

Indeed, we previously reported (Soh et al., 2015) that down-regulation of the *Stra8* transcript genetically depends on *Stra8,* but we did not show that this down-regulation is a direct result of STRA8 protein action.

In the present manuscript we have demonstrated that STRA8 acts primarily as a transcriptional activator. While it is possible that STRA8 could act as a transcriptional repressor at its own promoter, it is perhaps more likely that *Stra8-*dependent down-regulation of the *Stra8* gene is due to indirect negative effects of STRA8, such as a gene directly upregulated by STRA8 whose protein product can repress *Stra8* expression.

7) Explain the two categories of high and low Stra8: What is responsible for this variation? Please define the "low STRA8" are these spermatogonia, early pre-leptotene, or leptotene cells?

We have modified the second paragraph of the subsection “Transcriptional changes at the start of the meiotic cell cycle” to clarify the different categories of STRA8 expression. The “low-STRA8” cells are also in the preleptotene stage, but they express lower levels of STRA8 than “high-STRA8” preleptotene cells because they represent earlier (less developed) preleptotene cells.

8) The normalization procedure for RNA seq requires further investigation: it seems that RPMs have been used to normalize. Is this correct? What are the expression levels of housekeeping genes (wt vs. Stra8 KO)? Could these be used to normalize?Why is Rec8 down-regulated? Could this result from the normalization used?

We report our RNA-seq results in terms of transcripts per million (TPM), which is a newer but well-accepted method for RNA-seq normalization (see explanation at https://www.rna-seqblog.com/rpkm-fpkm-and-tpm-clearly-explained/). Within each sample, the TPM values of all genes sum to a million, which facilitates gene expression comparisons across many samples.

The observed *Rec8* down-regulation inhigh-STRA8 preleptotene cells compared to *Stra8-*deficient preleptotene cells is not due to a direct effect of STRA8 binding, butlikely reflects a down-regulation of *Rec8* following meiotic initiation. *Rec8* down-regulation was previously demonstrated in female meiosis using a FISH imaging method that requires no normalization; in the E14.5 embryonic ovary, *Rec8* transcript levels are lower in wild-type germ cells than in *Stra8-*deficient germ cells (Soh et al., 2015). These embryonic ovary results, which we have added to the third paragraph of the subsection “Transcriptional amplification of meiotic prophase factors at meiotic initiation”, suggest that proper meiotic initiation (which requires *Stra8*) results in the down-regulation of *Rec8* expression.

9) The authors should discuss what the down-regulated genes are and whether this is compatible with a direct and/or indirect effect.

As shown in Figure 3B, we find 165 STRA8-bound genes that are also down-regulated in the high-STRA8 preleptotene cells compared to *Stra8-*deficient cells. The significant depletion of STRA8-bound genes among down-regulated genes suggests that STRA8 does not act as a transcriptional repressor. However, we cannot formally exclude the possibility that STRA8 may directly repress the expression of some of these genes.

We do not find meaningful GO categories among the list of 2278 genes that are down-regulated at meiotic initiation (higher expression in *Stra8-*deficient preleptotene cells compared to high-STRA8 prelepotene cells) or among the list of 165 STRA8-bound and down-regulated genes.

10) Figure 1—figure supplement 2A: include the Stra8 KO control.

We have modified Figure 1—figure supplement 2A to include the *Stra8-*deficient control.

11) This sentence is not very convincing: “Even genes that were bound in only one replicate are likely true STRA8 targets; their promoters display appreciable ChIP-seq signal compared to non-target genes (Figure 2—figure supplement 4G).” What is the meaning of "appreciable"? Either revise or delete.

We have deleted the word “appreciable.”

12) What is the chromatin status of Stra8 bound promoters? A survey and analysis of available data for H3K4me3, H3K9ac, H3K27me3… should be performed. This could provide insight as to whether Stra8 is actually promoting initiation of transcription or release from pausing.

We are not aware of histone modification ChIP-seq datasets for purified germ cells directly before and after meiotic initiation that would allow relevant analyses to provide insight into STRA8’s mechanism of action.

13) In the Introduction, the authors mention the issue of the genetic background on the Stra8 mutant phenotype. Have they looked at whether some polymorphism between 129 and B6 could be detected at Stra8 binding sites? More generally what is the evolutionary conservation (across mammals and vertebrates) of Stra8 motifs at promoters?

We have now added Figure 6—figure supplement 1, which demonstrates the evolutionary conservation of STRA8 motifs. We show that in mouse, meiotic genes have a greater number of STRA8 motifs at their promoters than do non-meiotic genes, and that this enrichment of STRA8 motifs is conserved in human. We now discuss this motif conservation in the main text (subsection “Transcriptional amplification of meiotic prophase factors at meiotic initiation”).

We have not investigated polymorphisms between 129 and B6 at STRA8 binding sites.

14) Figure 1E needs a better presentation: the plot indicates 12545 genes, but the two boxes, pink and green, cover all of them and extend to a Log change of zero? Is it possible to visualize the 2278 and 2361 genes within this plot?

Figure 1E has been modified. The significantly down regulated genes (2278) are now colored pink, the significantly upregulated genes (2361) are colored green, and genes that are not significantly changed are colored gray.

15) Which database was used for the motifs bound by other transcription factors? The overlap with TCPAP2E is intriguing. What is the occurrence of TCPAP2E motifs over the promoters of Stra8 up and down-regulated genes?

We used the motif database in the HOMER motif discovery software (http://homer.ucsd.edu/homer/motif/motifDatabase.html), which includes motifs found in other databases.

The overlap of the STRA8 motif with the TFAP2E motif simply reflects a partial match within the STRA8 motif. We have added an extra supplementary figure (Figure 4—figure supplement 2), which demonstrates that canonical TFAP (AP-2) motifs are not enriched at STRA8 binding sites.